# DP-LDMs: Differentially Private Latent Diffusion Models

## Abstract

Diffusion models (DMs) are widely used for generating high-quality high-dimensional images in a non-differentially private manner. However, due to the notoriously slow training process of DMs, applying differential privacy (DP) to the training routine requires adding large amounts of noise, yielding poor-quality generated images. To address this challenge, recent papers suggest pre-training DMs with public data, then fine-tuning them with private data using DP-SGD for a relatively short period. In this paper, we further improve the current state of DMs with DP by adopting the *Latent* Diffusion Models (LDMs). LDMs are equipped with powerful pre-trained autoencoders that map the high-dimensional pixels into lower-dimensional latent representations, in which DMs are trained, yielding a more efficient and fast training of DMs. In our algorithm, DP-LDMs, rather than fine-tuning the entire DMs, we fine-tune only the *attention* modules of LDMs at varying layers with privacy-sensitive data, reducing the number of trainable parameters by roughly $90\%$ and achieving a better accuracy, compared to fine-tuning the entire DMs. The smaller parameter space to fine-tune with DP-SGD helps our algorithm to achieve new state-of-the-art results in several public-private benchmark data pairs. Our approach also allows us to generate more realistic, high-dimensional images (256x256) and those conditioned on text prompts with differential privacy, which have not been attempted before us, to the best of our knowledge. Our approach provides a promising direction for training more powerful, yet training-efficient differentially private DMs, producing high-quality high-dimensional DP images.

## 1 Introduction

Creating impactful machine learning solutions for real-world applications often requires access to personal data that may compromise privacy, raising ethical and legal concerns. These reasons motivate *differentially private data generation* as an active area of research. The main objective of this research is to generate synthetic data that preserves the privacy of the individuals in the original data while maintaining the statistical properties of the original data. Unlike traditional methods that require strict assumptions about the intended use of synthetic data (Mohammed et al., 2011; Xiao et al., 2010; Hardt et al., 2012; Zhu et al., 2017), recent approaches aim to create synthetic data that is general-purpose and useful for a range of downstream tasks, including training a classifier and performing statistical testing. These popular approaches include GAN-based models (Xie et al., 2018; Torkzadehmahani et al., 2019; Yoon et al., 2019; Chen et al., 2020), optimal transport or kernel-based distance approaches (Cao et al., 2021; Harder et al., 2021; Vinaroz et al., 2022; Yang et al., 2023), and diffusion models (Dockhorn et al., 2023; Ghalebikesabi et al., 2023).

Many of these popular approaches for differentially private data generation operate on small generative models, such as two-layer convolutional neural networks (CNNs), and simple datasets such as MNIST and FashionMNIST. This is because the DP training algorithm, called *differentially private stochastic gradient descent (DP-SGD)* by Abadi et al. (2016), does not scale well for large models that are necessary for learning complex distributions. For instance, the recent approach by Dockhorn et al. (2023) uses diffusion models that have shown impressive performance on high-dimensional image generation in non-DP settings. However, due to the scalability issue of DP-SGD, DP trained diffusion models yield rather underwhelming performance when evaluated on complex datasets such as CIFAR10 and CelebA.

More recent approaches attempt to overcome this issue by utilizing the abundant resource of *public* data. For example, Harder et al. (2023) use public data for pre-training a large feature extractor model to learn useful features without incurring a privacy loss, then use those features to train a generator using private data. As another example, Ghalebikesabi et al. (2023) pretrain a large diffusion-based generator using public data, then fine-tune it for private data for a relatively small number of epochs using DP-SGD. This method currently achieves the state-of-the-art performance on CIFAR10 image generation using DMs with differential privacy.

In this paper, we attempt to further improve the performance of differentially private image generation by reducing the number of parameters to fine-tune. To achieve this, we build off of *latent diffusion models (LDMs)* by Rombach et al. (2022), which uses a pre-trained autoencoder to reduce the size of images, often called *latent variables*, entering into the diffusion model. This latent diffusion model defined on this latent space has a significantly lower number of parameters than the diffusion model defined on the pixel space. Inspired by You & Zhao (2023) that establishes a transfer learning paradigm for LDMs in non-DP settings, we pre-train the entire LDM including the auto-encoder using public data, and fine-tune only attention modules and a conditioning embedder using our private data. As a result, the number of trainable parameters under our approach is only $10\%$ of that of the diffusion models used in (Ghalebikesabi et al., 2023) and achieves better performance.

While readers might find the use of DP-SGD to fine-tune a pre-trained model unremarkable at first glance, the potential impact of this seemingly ordinary method is substantial. We describe the significance in the following:

- *We improve the state-of-the-art (SOTA) results in all three commonly used image benchmark datasets in DP literature, including CIFAR10, CelebA64, and MNIST.* This is thanks to the unique aspects of our proposed method, i.e., training DMs in the latent space and fine-tuning only a few selected parameters. This makes our training process considerably more efficient than training a DM from scratch with DP-SGD in (Dockhorn et al., 2023), or fine-tuning the entire DM with DP-SGD in (Ghalebikesabi et al., 2023). Reducing the fine-tuning space not only improves the performance but also helps to democratize DP image generation using diffusion models, which otherwise have to rely on massive computational resources only available to a small fraction of the field and would leave a huge carbon footprint (e.g., reducing the training time from 192 GPU hours Dockhorn et al. (2023) to mere 10 GPU hours for similar performance).

- *We push the boundary of what DP-SGD fine-tuned generative models can achieve*, by being the first to produce high-dimensional images (256x256) at a reasonable privacy level. We showcase this in text-conditioned and class-conditioned image generation, where we input a certain text prompt (or a class label) and generate a corresponding image from a DP-fine-tuned LDM for CelebAHQ. These conditional, high-dimensional image generation tasks present more complex but also more realistic benchmarks compared to the conventional CIFAR10 and MNIST datasets. These latter datasets, though widely used in DP image generation literature for years, are now rather simplistic and outdated. Our work contributes to narrowing down the large gap between the current state of synthetic image generation in non-DP settings and that in DP settings.

In the following section, we provide relevant background information. We then present our method along with related work and experiments on benchmark datasets.

## 2 BACKGROUND

We first describe latent diffusion models (LDMs), then the definition of differential privacy (DP) and finally the DP-SGD algorithm, which we will use to train the LDM in our method.

### 2.1 LATENT DIFFUSION MODELS (LDMS)

Diffusion Models gradually denoise a normally distributed variable through learning the reverse direction of a Markov Chain of length $T$. Latent diffusion models (LDMs) by Rombach et al. (2022) are a modification of denoising diffusion probabilistic models (DDPMs) by Ho et al. (2020) in the following way. First, Rombach et al. (2022) uses a powerful auto-encoder, consisting of an

encoder Enc and a decoder Dec . The encoder transforms a high-dimensional pixel representation $\mathbf{x}$ into a lower-dimensional latent representation $\mathbf{z}$ via $\mathbf{z} = \text{Enc}(\mathbf{x})$; and the decoder transforms the lower-dimensional latent representation back to the original space via $\hat{\mathbf{x}} = \text{Dec}(\mathbf{z})$. Rombach et al. (2022) use a combination of a perceptual loss and a patch-based adversarial objective, with extra regularization for better-controlled variance in the learned latent space, to obtain powerful autoencoders (See section 3 in (Rombach et al., 2022) for details). This training loss encourages the latent representations to carry equivalent information (e.g., the spatial structure of pixels) as the pixel representations, although the dimensionality of the former is greatly reduced.

Second, equipped with the powerful auto-encoder, Rombach et al. (2022) trains a diffusion model (typically a UNet (Ronneberger et al., 2015)) in the latent representation space. Training a DM in this space can significantly expedite the training process of diffusion models, e.g., from hundreds of GPU *days* to several GPU *hours* for similar accuracy.

Third, LDMs also contain *attention modules* (Vaswani et al., 2017) that take inputs from a *conditioning embedder*, inserted into the layers of the underlying UNet backbone as the way illustrated in Fig. 1 to achieve flexible conditional image generation (e.g., generating images conditioning on text, image layout, class labels, etc.). The modified Unet is then used as a function approximator $\tau_{\boldsymbol{\theta}}$ to predict an initial noise from the noisy lower-dimensional latent representations at several finite time steps $t$, where in LDMs, the noisy representations (rather than data) follow the diffusion process defined in Ho et al. (2020).

The parameters of the approximator are denoted by $\boldsymbol{\theta} = [\boldsymbol{\theta}^U, \boldsymbol{\theta}^{Att}, \boldsymbol{\theta}^{Cn}]$, where $\boldsymbol{\theta}^U$ are the parameters of the underlying Unet backbone, $\boldsymbol{\theta}^{Att}$ are the parameters of the attention modules, and $\boldsymbol{\theta}^{Cn}$ are the parameters of the conditioning embedder (We will explain these further in Sec. 3). These parameters are then optimized by minimizing the prediction error defined by

$$\mathcal{L}_{ldm}(\boldsymbol{\theta}) = \mathbb{E}_{(\mathbf{z}_t, y), \tau, t} \left[ \| \tau - \tau_{\boldsymbol{\theta}}(\mathbf{z}_t, t, y) \|_2^2 \right], \tag{1}$$

where $\tau \sim \mathcal{N}(0, I)$, $t$ uniformly sampled from $\{1, \cdots, T\}$, $x_t$ is the noisy version of the input $x$ at step $t$, $\mathbf{z}_t = \text{Enc}(\mathbf{x}_t)$ and $y$ is what the model is conditioning on to generate data, e.g., class labels, or a prompt. Once the approximator is trained, the drawn samples in latent space, $\tilde{\mathbf{z}}$, are transformed back to pixel space through the decoder, i.e., $\tilde{\mathbf{x}} = \text{Dec}(\tilde{\mathbf{z}})$. Our work introduced in Sec. 3 pre-trains both auto-encoder and $\tau_{\boldsymbol{\theta}}$ using public data, then fine-tunes only $\boldsymbol{\theta}_{Att}, \boldsymbol{\theta}_{Cn}$, the parameters of the attention modules and the conditioning embedder, using DP-SGD for private data.

## 2.2 DIFFERENTIAL PRIVACY (DP)

A mechanism $\mathcal{M}$ is $(\epsilon, \delta)$-DP for a given $\epsilon \geq 0$ and $\delta \geq 0$ if and only if $\Pr[\mathcal{M}(\mathcal{D}) \in S] \leq e^{\epsilon} \cdot \Pr[\mathcal{M}(\mathcal{D}') \in S] + \delta$ for all possible sets of the mechanism's outputs $S$ and all neighbouring datasets $\mathcal{D}, \mathcal{D}'$ that differ by a single entry. A single entry difference could come from either replacing or removing one entry from the dataset $\mathcal{D}$. One of the most well-known and widely used DP mechanisms is the *Gaussian mechanism*. The Gaussian mechanism adds a calibrated level of noise to a function $\mu : \mathcal{D} \mapsto \mathbb{R}^p$ to ensure that the output of the mechanism is $(\epsilon, \delta)$-DP: $\widetilde{\mu}(\mathcal{D}) = \mu(\mathcal{D}) + n$, where $n \sim \mathcal{N}(0, \sigma^2 \Delta_{\mu}^2 \mathbf{I}_p)$. Here, $\sigma$ is often called a privacy parameter, which is a function of $\epsilon$ and $\delta$. $\Delta_{\mu}$ is often called the *global sensitivity* (Dwork et al., 2006; 2014), which is the maximum difference in $L_2$-norm given two neighbouring $\mathcal{D}$ and $\mathcal{D}'$, $||\mu(\mathcal{D}) - \mu(\mathcal{D}')||_2$. Because we are adding noise, the natural consequence is that the released function $\tilde{\mu}(\mathcal{D})$ is less accurate than the non-DP counterpart, $\mu(\mathcal{D})$. This introduces privacy-accuracy trade-offs.

Two properties of DP: The *post-processing invariance* property of DP (Dwork et al., 2006; 2014) implies that the composition of any data-independent mapping with an $(\epsilon, \delta)$-DP algorithm is also $(\epsilon, \delta)$-DP. So no analysis of the released synthetic data can yield more information about the real data than what our choice of $\epsilon$ and $\delta$ allows. The *composability* property Dwork et al. (2006; 2014) states that the strength of privacy guarantee degrades in a measurable way with repeated use of DP-algorithms. This composability property of DP poses a significant challenge in deep learning.

DP-SGD (Abadi et al., 2016) is one of the most widely used DP algorithms for training deep neural network models. It modifies stochastic gradient descent (SGD) by adding an appropriate amount of noise to the gradients in every training step, where the amount of noise is controlled by Gaussian mechanism. This adjustment ensures the parameters of a neural network are differentially private. However, there are two challenges in DP-SGD. First, it is infeasible to obtain an analytic sensitivity of

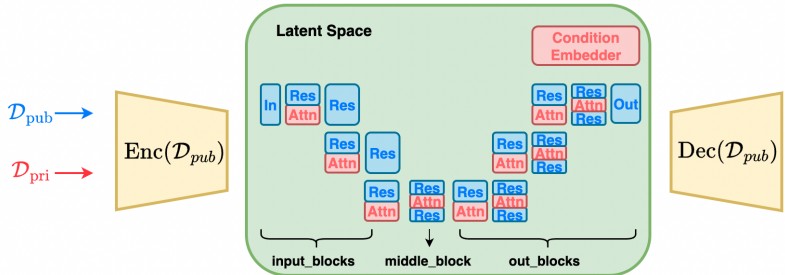

Figure 1: A schematic of DP-LDM. In the non-private step, we pre-train the auto-encoder depicted in yellow (Right and Left) with public data. We then forward pass the public data through the encoder (blue arrow on the left) to obtain latent representations. We then train the diffusion model (depicted in the green box) on the lower-dimensional latent representations. The diffusion model consists of the Unet backbone and added attention modules (in Red) with a conditioning embedder (in Red, at top-right corner). In the private step, we forward pass the private data (red arrow on the left) through the encoder to obtain latent representations of the private data. We then fine-tune only the red blocks, which are attention modules and conditioning embedder, with DP-SGD. Once the training is done, we sample the latent representations from the diffusion model, and pass them through the decoder to obtain the image samples in the pixel space.

gradients under complex deep neural network models. A remedy to this issue is explicitly normalizing the norm of each sample-wise gradient with some pre-chosen value $C$ such that the gradient norm given any datapoint's difference between two neighbouring datasets cannot exceed $C$. Second, due to the composability property of DP, privacy loss is accumulating over a typically long course of training. Abadi et al. (2016) exploit the subsampled Gaussian mechanism (i.e., applying the Gaussian mechanism on randomly subsampled data) to achieve a tight privacy bound. The *Opacus* package (Yousefpour et al., 2021) implements the privacy analysis in DP-SGD, which we adopt in our method. One thing to note is that we use the **removing** definition of DP in the experiments as in *Opacus*.

## 3 DIFFERENTIALLY PRIVATE LATENT DIFFUSION MODELS (DP-LDMs)

In our method, which we call *differentially private latent diffusion models (DP-LDMs)*, we carry out two training steps: non-private and private steps.

**Non-Private Step: Pre-training an autoencoder and a DM using public data.** Following Rombach et al. (2022), we first pre-train an auto-encoder. The encoder scales down an image $\mathbf{x} \in \mathbb{R}^{H \times W \times 3}$ to a 3-dimensional latent representation $\mathbf{z} \in \mathbb{R}^{h \times w \times c}$ by a factor of $f$, where $f = H/h = W/w$. This 3-dimensional latent representation is chosen to take advantage of image-specific inductive biases that the Unet contains, e.g., 2D convolutional layers. Following Rombach et al. (2022), we also train the autoencoder by minimizing a combination of different losses, such as perceptual loss and adversarial loss, with some form of regularization. See Appendix Sec. A.1 for details. As noted by Rombach et al. (2022), we also observe that a mild form of downsampling performs the best, achieving a good balance between training efficiency and perceptually decent results. See Appendix Sec. A.1 for details on different scaling factors $f = 2^m$, with a different value of $m$. Training an auto-encoder does not incur any privacy loss, as we use public data $\mathcal{D}_{pub}$ that is similar to private data $\mathcal{D}_{priv}$ at hand. The trained autoencoder is, therefore, a function of public data: an encoder $\mathrm{Enc}(\mathcal{D}_{pub})$ and a decoder $\mathrm{Dec}(\mathcal{D}_{pub})$.

A forward pass through the trained encoder $\mathrm{Enc}(\mathcal{D}_{pub})$ gives us a latent representation of each image, which we use to train a diffusion model. As in (Rombach et al., 2022), we consider a modified Unet for the function approximator $\tau_{\boldsymbol{\theta}}$ shown in Fig. 1. We minimize the loss given in (1) to estimate the parameters of $\tau_{\boldsymbol{\theta}}$ as:

$$\boldsymbol{\theta}^U_{\mathcal{D}_{pub}}, \boldsymbol{\theta}^{Att}_{\mathcal{D}_{pub}}, \boldsymbol{\theta}^{Cn}_{\mathcal{D}_{pub}} = \arg\min_{\boldsymbol{\theta}} \ \mathcal{L}_{ldm}(\boldsymbol{\theta}). \tag{2}$$

Since we use public data, there is no privacy loss incurred in estimating the parameters, which are a function of public data $\mathcal{D}_{pub}$.

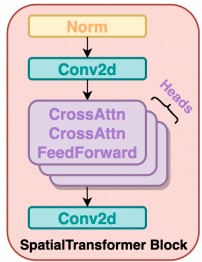

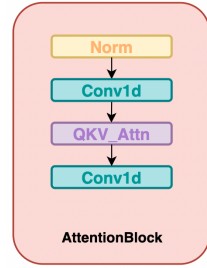

Figure 2: A SpatialTransformer Block      Figure 3: An AttentionBlock

---

**Algorithm 1** DP-LDM

**Input:** Latent representations through a pre-trained auto-encoder and conditions (if conditioned generation) $\{(\mathbf{z}_i, y_i)\}_{i=1}^N$, a pre-trained diffusion model with parameters $\boldsymbol{\theta}$, number of iterations $P$, mini-batch size $B$, clipping norm $C$, learning rate $\eta$, privacy parameter $\sigma$ corresponding to ($\epsilon$, $\delta$)-DP. Denote $\hat{\boldsymbol{\theta}} = \{\boldsymbol{\theta}^{Att}, \boldsymbol{\theta}^{Cn}\}$

**for** $p = 1$ **to** $P$ **do**

    **Step 1**. Take a mini-batch $B_p$ uniformly at random with a sampling probability, $q = B/N$

    **Step 2**. For each sample $i \in B_p$ compute the gradient: $g_p(\mathbf{z}_i, y_i) = \nabla_{\hat{\boldsymbol{\theta}}_p} \mathcal{L}_{ldm}(\hat{\boldsymbol{\theta}}_p, \mathbf{z}_i, y_i)$

    **Step 3**. Clip the gradient: $\hat{g}_p(\mathbf{z}_i, y_i) = g_p(\mathbf{z}_i, y_i) / \max(1, \|g_p(\mathbf{z}_i, y_i)\|_2 / C)$

    **Step 4**. Add noise: $\tilde{g}_p = \frac{1}{B} \left( \sum_{i=1}^B \hat{g}_p(\mathbf{z}_i, y_i) + \mathcal{N}(0, \sigma^2 C^2 I) \right)$

    **Step 5**. Gradient descent: $\hat{\boldsymbol{\theta}}_{p+1} = \hat{\boldsymbol{\theta}}_p - \eta \tilde{g}_p$

**end for**

**Return:** ($\epsilon$, $\delta$)-differentially private $\hat{\boldsymbol{\theta}}_P = \{\boldsymbol{\theta}_P^{Att}, \boldsymbol{\theta}_P^{Cn}\}$

---

**Private Step: Fine-tuning attention modules & conditioning embedder for private data.** Given a pre-trained diffusion model, we fine-tune the attention modules and a conditioning embedder using our private data. For the models with the conditioned generation, the attention modules refer to the spatial transformer blocks shown in Fig. 2 which contains cross-attentions and multiple heads. For the models with an unconditional generation, the attention modules refer to the attention blocks shown in Fig. 3. Consequently, the parameters of the attention modules, denoted by $\boldsymbol{\theta}^{Att}$, differ, depending on the conditioned or unconditioned cases. The conditioning embedder only exists in the conditioned case. Depending on the different modalities the model is trained on, the conditioning embedder takes a different form. For instance, if the model generates images conditioning on the class labels, the conditioning embedder is simply a class embedder, which embeds class labels to a latent dimension. If the model conditions on language prompts, the embedder can be a transformer.

The core part of the spatial transformer block and the attention block is the attention layer, which has the following parameterization (For simplicity, we explain it under the conditioned case):

$$\text{Attention}(\psi_i(\mathbf{z}_t), \phi(y); Q, K, V) = \text{softmax}\left(\frac{QK^T}{\sqrt{d_k}}\right) V \ \in \mathbb{R}^{N \times d_k}, \tag{3}$$

where $\psi_i(\mathbf{z}_t) \in \mathbb{R}^{N \times d^i}$ is an intermediate representation of the latent representation $\mathbf{z}_t$ through the $i$th residual convolutional block in the backbone Unet, and $\phi(y) \in \mathbb{R}^{M \times d_c}$ is the embedding of what the generation is conditioned on (e.g., class labels, CLIP embedding). Furthermore, $Q = \psi_i(\mathbf{z}_t)W_Q^{(i)\top}$, $K = \phi(y)W_K^{(i)\top}$, and $V = \phi(y)W_V^{(i)\top}$, where the parameters are denoted by $W_Q^{(i)} \in \mathbb{R}^{d_k \times d^i}$; $W_K^{(i)} \in \mathbb{R}^{d_k \times d_c}$; and $W_V^{(i)} \in \mathbb{R}^{d_k \times d_c}$. Unlike the conditioned case, where the key ($K$) and value ($V$) vectors are computed as a projection of the conditioning embedder, the key and value vectors are a projection of the pixel embedding $\psi_i(\mathbf{z}_t)$ only in the case of the unconditioned model. We run DP-SGD to fine-tune these parameters to obtain differentially private $\boldsymbol{\theta}_{\mathcal{D}_{priv}}^{Att}$ and $\boldsymbol{\theta}_{\mathcal{D}_{priv}}^{Cn}$, starting from $\boldsymbol{\theta}_{\mathcal{D}_{pub}}^{Att}, \boldsymbol{\theta}_{\mathcal{D}_{pub}}^{Cn}$. Our algorithm is given in Algorithm 1.

**Why do we choose the attention modules to be fine-tuned among any other parts of the model?** Our rationale behind this choice is as follows. The output of the attention in (3) assigns a high focus

to the features that are more important, by zooming into what truly matters in an image depending on a particular context, e.g., relevant to what the image is conditioned on. This can be quite different when we move from one distribution to the other. By fine-tuning the attention modules (together with the conditioning embedder when conditioned case), we are able to effectively transfer what we learned from the public data distribution to the private data distribution, as shown in Sec. 5. However, if we fine-tune other parts of the model, e.g., the ResBlocks, the fine-tuning of these blocks can make a large change in the features themselves, which could reduce the performance in the private training. See our results when fine-tuning other parts of the model in Sec. 5.

The idea of fine-tuning attention blocks is explored elsewhere. For instance, in fine-tuning large language models, existing work introduces a few new parameters to transformer attention blocks, and those new parameters are fine-tuned (Yu et al., 2022; Hu et al., 2021) to adapt to new distributions. In the context of pre-trained diffusion models, adding, modifying, and controlling attention layers are gaining popularity for tasks such as image editing and text-to-image generation(Hertz et al., 2022; Park et al., 2023; Zhang et al., 2023; You & Zhao, 2023).

**Which public dataset do I use for a given private dataset?** This is an open question in transfer learning literature. Generally, if the two datasets are close to each other in some sense, they are assumed to be a better pair. We use FID as a proxy to judge the similarity between two image datasets. For instance, if a public dataset from the private dataset has a smaller FID than other candidates, we use that public data to begin with (See Sec. 5). In other datasets out of the image domain, there could be a more appropriate metric to use than FID, e.g., in the case of discrete data, kernel-based distance metrics with an appropriately chosen kernel could be more useful.

## 4 RELATED WORK

Early efforts in differentially private data generation imposes strict limitations on the data type and the intended purpose of the released data (Snoke & Slavković, 2018; Mohammed et al., 2011; Xiao et al., 2010; Hardt et al., 2012; Zhu et al., 2017), which leads to the difficulties in generating large-scale data. Later, several works have explored generating discrete data with restricted range of values, by understanding the relationships of small groups of features and then privatizing them (Zhang et al., 2017; Qardaji et al., 2014; Chen et al., 2015; Zhang et al., 2021). However, these techniques cannot be applied to high-dimensional data due to the constraint of discretization. Recently, more efforts have focused on leveraging advanced generative models to achieve better differentially private synthetic data (Hu et al., 2023). Some of them (Xie et al., 2018; Torkzadehmahani et al., 2019; Frigerio et al., 2019; Yoon et al., 2019; Chen et al., 2020) utilize generative adversarial networks (GANS) (Goodfellow et al., 2014), or trained GANs with the PATE structure (Papernot et al., 2017). Other works have employed variational autoencoders (VAEs) (Acs et al., 2018; Jiang et al., 2022; Pfitzner & Arnrich, 2022), or proposed customized structures (Harder et al., 2021; Vinaroz et al., 2022; Cao et al., 2021; Liew et al., 2022a; Harder et al., 2023). For instance, Harder et al. (2023) pretrained perceptual features using public data and privatized only data-dependent terms using maximum mean discrepancy.

Limited works have so far delved into privatizing diffusion models. Dockhorn et al. (2023) develop a DP score-based generative models Song et al. (2021) using DP-SGD, applied to relatively simple datasets such as MNIST, FashionMNIST and CelebA (downsampled to $32\times32$). Ghalebikesabi et al. (2023) fine-tune the ImageNet pre-trained diffusion model (DDPM) (Ho et al., 2020) with more than 80 M parameters using DP-SGD for CIFAR-10. We instead adopt a different model (LDM) and fine-tune only the small part of the DM in our model to achieve better privacy-accuracy trade-offs. As concurrent work to ours, Lin et al. (2023) propose an API-based approach that uses a DP-histogram mechanism to generate high-quality synthetic data. However, Lin et al. (2023) do not privatize diffusion models directly, so we do not compare our method against it.

## 5 EXPERIMENTS

Here, we demonstrate the performance of our method in comparison with the state-of-the-art methods in DP data generation, using several combinations of public/private data of different levels of complexity at varying privacy levels.

Table 1: FID scores (lower is better) for synthetic CIFAR-10 data.

|  | $\epsilon = 10$ | $\epsilon = 5$ | $\epsilon = 1$ |
|---|---|---|---|
| **DP-LDM** | **8.4 $\pm$ 0.2** | **13.4 $\pm$ 0.4** | **22.9 $\pm$ 0.5** |
| DP-Diffusion | 9.8 | 15.1 | 25.2 |
| DP-MEPF ($\phi_1, \phi_2$) | 29.1 | 30.0 | 54.0 |
| DP-MEPF ($\phi_1$) | 30.3 | 35.6 | 56.5 |

Table 2: Test accuracies (higher is better) of ResNet9 (left) and WRN40-4 (right) trained on CIFAR-10 synthetic data. When trained on real data, test accuracy is 88.3% on ResNet9.

| | ResNet 9 | | | | WRN40-4 | |
|---|---|---|---|---|---|---|
| | $\epsilon = 10$ | $\epsilon = 5$ | $\epsilon = 1$ | | | $\epsilon = 10$ |
| **DP-LDM** | **65.3 $\pm$ 0.3** | **59.1 $\pm$ 0.2** | **51.3 $\pm$ 0.1** | | **DP-LDM** | **78.6 $\pm$ 0.3** |
| DP-MEPF ($\phi_1, \phi_2$) | 48.9 | 47.9 | 28.9 | | DP-Diffusion | 75. 6 |
| DP-MEPF ($\phi_1$) | 51.0 | 48.5 | 29.4 | | | |
| DP-MERF | 13.2 | 13.4 | 13.8 | | | |

**Datasets.** For private datasets, we considered three image datasets[1] of varying complexity: the commonly used datasets MNIST (LeCun & Cortes, 2010), the more complex datasets CelebA (Liu et al., 2015) , and also CIFAR-10 (Krizhevsky et al., 2009). In addition, we also consider the high-quality version CelebAHQ (Karras et al., 2018) of size $256 \times 256$ to generate high-dimensional images. For text-to-image generation, we used Multi-Modal-CelebAHQ (MM-CelebAHQ)(Xia et al., 2021) dataset, containing of 30,000 $256 \times 256$ images, each of which is accompanied by descriptive text captions. As public datasets, we used EMNIST (Cohen et al., 2017) English letter split parts for MNIST, ImageNet (Deng et al., 2009) (rescaled it to the corresponding sizes) for CelebA and CIFAR-10 and CelebAHQ, and LAION-400M (Schuhmann et al., 2021) for MM-CelebAHQ.

**Implementations.** We implemented DP-LDMs in PyTorch Lightning (Paszke et al., 2019) building on the LDM codebase by Rombach et al. (2022) and Opacus (Yousefpour et al., 2021) for DP-SGD training. Several recent papers present the importance of using large batches in DP-SGD training to improve accuracy at a fixed privacy level (Ponomareva et al., 2023; De et al., 2022; Bu et al., 2022). To incorporate this finding in our work, we write custom batch splitting code that integrates with Opacus and Pytorch Lightning, allowing us to test arbitrary batch sizes. Our DP-LDM also improves significantly with large batches as will be shown soon, consistent with the findings in the recent work.

**Evaluation.** For the simple dataset MNIST, we take generated samples to train downstream classifiers and compute the accuracy on real data. We consider CNN and MLP classifiers as in standard practice, and also Wide ResNet (WRN-40-4), a much larger classifier. For CelebA, CIFAR-10, CelebAHQ, and MM-CelebAHQ, we measure the model performance by computing the Fréchet Inception Distance (FID) (Heusel et al., 2017) between the generated samples and the real data. For CIFAR-10, we additionally train downstream classifiers (ResNet-9 and WRN-40-4) to compare against other state-of-the-art methods. Each number in our tables represents an average value across three independent runs, with a standard deviation (unless stated otherwise). Values for comparison methods are taken from their papers, with an exception for the DP-MEPF comparison to CelebAHQ, which we ran their code by loading this data.

## 5.1 TRANSFERRING FROM IMAGENET TO CIFAR10 DISTRIBUTION

First, we pre-train a class-conditional LDM model considering ImageNet32 as public data. The Unet we use has 16 SpatialTransformer blocks as in Fig. 2. For the fine-tuning part, we consider CIFAR-10 as the private dataset and test the performance of DP-LDM at different privacy levels for 3 independent sampling runs. See Appendix Sec. A.3 for all experimental details.

Comparison to other SOTA methods in terms of FID (the lower the better) is given in Table 1. Our DP-LDM outperforms other methods at all epsilon levels ($\epsilon = 1, 5, 10$ and $\delta = 10^{-5}$). These FID values correspond to the case where only 9-16 attention modules are fine-tuned (i.e., fine-tuning

---

[1]Dataset licenses: MNIST: CC BY-SA 3.0; CelebA: see `https://mmlab.ie.cuhk.edu.hk/projects/CelebA.html`; Cifar10: MIT

Table 3: Fine-tuning different parts of the model for synthetic CIFAR-10 images.

| | $\epsilon = 1$ | | | $\epsilon = 10$ | |
| | Input block | Attention module | | Resblocks | Attention module |
|---|---|---|---|---|---|
| FID | $50.2 \pm 0.2$ | $\mathbf{22.9 \pm 0.5}$ | FID | $86.9 \pm 0.6$ | $\mathbf{8.4 \pm 0.2}$ |

Table 4: CelebA FID scores (lower is better) for images of resolution $64 \times 64$.

| | $\epsilon = 10$ | $\epsilon = 5$ | $\epsilon = 1$ |
|---|---|---|---|
| DP-LDM (ours, average) | $14.3 \pm 0.1$ | $16.1 \pm 0.2$ | $21.1 \pm 0.2$ |
| DP-LDM (ours, best) | $\mathbf{14.2}$ | $\mathbf{15.8}$ | $21.0$ |
| DP-MEPF ($\phi_1$) | $17.4$ | $16.5$ | $\mathbf{20.4}$ |
| DP-GAN (pre-trained) | $57.1$ | $62.3$ | $72.5$ |

only $10\%$ of trainable parameters in the model) and the rest remain fixed. See Table 10 for ablation experiments for fine-tuning different attention modules. In terms of downstream accuracy of ResNet-9 and WRN40-4 classifiers, DP-LDM also outperforms others, as shown in Table 2. Both classifiers are trained with 50K synthetic samples and then evaluated on real data samples.

When we fine-tune other parts of the model, we see a significant drop in accuracy as shown in Table 3. This confirms that fine-tuning resblocks or input blocks alone is less effective than fine-tuning attention modules at a fixed privacy budget.

## 5.2 TRANSFERRING FROM IMAGENET TO CELEBA DISTRIBUTION

We evaluate the performance of our model on the task of unconditional image generation for CelebA (rescaled to 64x64) using an LDM pretrained on ImageNet. Additional experiments for CelebA32 are available in appendix A.4 We compare DP-LDM to existing methods at the privacy settings $\delta = 10^{-6}$ and $\epsilon = 1, 5, 10$ in Table 4. We achieve a new SOTA results at $\epsilon = 10, 5$ and are comparable to DP-MEPF at $\epsilon = 1$. Samples are available in Figure 9. Table 5 provides evidence suggesting that training with larger batch sizes improves the performance of the model.

Table 5: Effect of increasing batch size on FID. At a fixed epsilon level, larger batches improve FIDs.

| Batch size | $\epsilon = 10$ | $\epsilon = 5$ | $\epsilon = 1$ |
|---|---|---|---|
| 512 | $17.2 \pm 0.1$ | $18.0 \pm 0.1$ | $22.3 \pm 0.2$ |
| 2048 | $16.2 \pm 0.2$ | $17.1 \pm 0.2$ | $22.1 \pm 0.1$ |
| 8192 | $\mathbf{14.3 \pm 0.1}$ | $\mathbf{16.1 \pm 0.2}$ | $\mathbf{21.1 \pm 0.2}$ |

## 5.3 TRANSFERRING FROM EMNIST TO MNIST DISTRIBUTION

To choose the public dataset for MNIST, we consider SVHN (Netzer et al., 2011), KMNIST (Clanuwat et al., 2018), and EMNIST (Cohen et al., 2017) as candidates. As a selection criterion, we computed the FID score between the samples from each candidate dataset and those from MNIST. Based on the FID scores (See Appendix Sec. A.2.1 for FIDs and additional experiments for SVHN and KMNIST), we chose EMNIST as the public dataset to pretrain a class-conditional LDM. We only took the English letter split of EMNIST such that the model does not see the MNIST data during the non-private pre-training step. We then finetune the attention modules with MNIST and test the performance of DP-LDM with downstream classifiers. We compare DP-LDM to existing methods with the most common DP settings $\epsilon = 1, 10$ and $\delta = 10^{-5}$ in Table 7 in Appendix Sec. A.2. We also did ablation experiments to reduce the trainable parameters in Appendix Sec. A.2.2.

## 5.4 DIFFERENTIALLY PRIVATE GENERATION FOR HIGH QUALITY CELEBA

With the latent representations of the inputs, LDMs can better improve DP training. We present high-dimensional differentially private generation for CelebAHQ in this section.

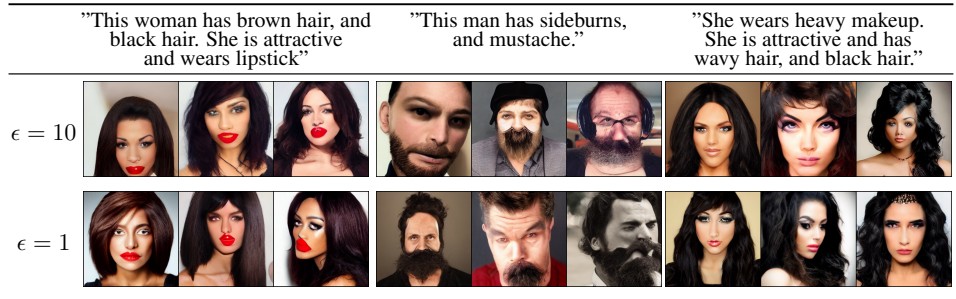

Figure 4: Text-to-image generation for CelebAHQ 256 by 256 with three sample prompt inputs. Privacy condition is set to $\epsilon = 10, 1$ and $\delta = 10^{-5}$.

DP-LDM (Ours)                                                    DP-MEPF

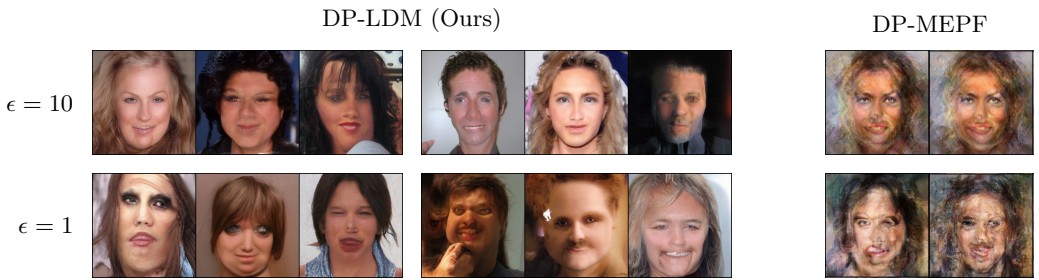

Figure 5: Synthetic $256 \times 256$ CelebA samples generated at different levels of privacy. Samples for DP-MEPF are generated from code available in Harder et al. (2023). We computed FID between our generated samples and the real data and achieve FIDs of $19.0 \pm 0.0$ at $\epsilon = 10$, $20.5 \pm 0.1$ at $\epsilon = 5$, and $25.6 \pm 0.1$ at $\epsilon = 1$. DP-MEPF achieves an FID of $200.8$ at $\epsilon = 10$ and $293.3$ at $\epsilon = 1$.

**Text-to-image generation.** For text-to-image generation, we fine-tune the LDM models pretrained with LAION-400M (Schuhmann et al., 2021) for MM-CelebAHQ ($256 \times 256$). Each image is described by a caption, which is fed to the conditioning embedder, *BERT* (Devlin et al., 2018). We freeze the BERT embedder during fine-tuning to reduce the trainable parameters, then we bring back BERT for sampling. DP-LDM achieves FID scores of $44.5$ for $\epsilon = 10$ and $55.0$ for $\epsilon = 1$. We illustrate our samples with example prompts in Fig. 5.4. The samples are faithful to our input prompts even at the $\epsilon = 1$ level.

**Class conditional generation.** We build our model on the LDM model provided by Rombach et al. (2022) which is pretrained on Imagenet at a resolution of $256 \times 256$. Following our experiments in Section 5.2, we fine-tune all of the SpatialTransformer blocks. While CelebAHQ does not provide class labels, each image is associated with 40 binary attributes. We choose the attribute "Male" to act as a binary class label for each image. Generated samples are available in Figure 5 along with FID values. Compared to DP-MEPF, based on the FID scores and perceptual image quality, DP-LDM is better suited for generating detailed, plausible samples from the high-resolution dataset at a wide range of privacy levels.

## 6 CONCLUSION AND DISCUSSION

In *Differentially Private Latent Diffusion Models* (DP-LDM), we utilize DP-SGD to finetune only the attention modules (and embedders for conditioned generation) of the pretrained LDM at varying layers with privacy-sensitive data. We demonstrate that our method is capable of generating images for simple datasets like MNIST, more complex datasets like CIFAR-10 and CelebA, and high-dimensional datasets like CelebAHQ. We perform an in-depth analysis of ablation of DP-LDM to explore the strategy to reducing parameters for more applicable training of DP-SGD. Based on our promising results, we conclude that fine-tuning LDMs is an efficient and effective framework for DP generative learning. We hope our results can contribute to future research in DP data generation, considering the rapid advances in diffusion-based generative modelling.

## 7 ETHICS AND REPRODUCIBILITY

As investigated in (Carlini et al., 2023), diffusion models can memorize individual images from the training data and gives the same as generating samples. Aiming to bringing positive effects to society, our research is driven by the necessity of robust and scalabel data privacy. However, it is also important to approach the use of public data cautiously. As (Tramèr et al., 2022) pointed out, public data themselves may still be sensitive due to lack of curation practices. In addition, the public data usually achieve similar distribution as the private data, however, no proper public data is available currently as this might require heavy domain shift of the data themselves. We understand these potential issues but current DP machine learning research leads to minimal effects because of the inadequacy of the utility. From our perspective, auxiliary public data still emerges as the most promising option for attaining satisfactory utility, comparing to the potential harm it might inject. We hope our discussion will contribute to further research in differential privacy in machine learning using public data.

To guarantee the reproduciblity, our code is available at `https://anonymous.4open.science/r/DP-LDM-4525` with detailed instructions. And all the hyperparameters are discussed in detail in Appendix Sec. B.

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

# Appendix

## A ADDITIONAL EXPERIMENTS

### A.1 SCALING FACTOR EFFECT IN PRE-TRAINING THE AUTOENCODER

In Table 6, we provide FID results after pre-training the autoencoder with Imagenet dataset for different scaling factors $f$ and number of channels.

Table 6: FID scores (lower is better) for pre-trained autoencoders with different $f$ and number of channels.

|          | # channels |      |      |
| -------- | ---------- | ---- | ---- |
|          | 128        | 64   | 32   |
| $f = 2$  | **27.6**   | 36.4 | 46.8 |
| $f = 4$  | 32.9       | 51.0 | 83.5 |

### A.2 TRANSFERRING FROM EMNIST TO MNIST DISTRIBUTION

Here we compare DP-LDM to existing methods with the most common DP settings $\epsilon = 1, 10$ and $\delta = 10^{-5}$ in Table 7.

Table 7: Downstream accuracies by CNN, MLP and WRN-40-4, evaluated on the generated MNIST data samples. We compare our results with existing work DPDM (Dockhorn et al., 2023), DP-Diffusion (Ghalebikesabi et al., 2023), PEARL (Liew et al., 2022b), DPGANr (Bie et al., 2022), and DP-HP (Vinaroz et al., 2022). The GPU hours is for DP training only. The GPU hours for pretraining steps of our method are present in Table 14 and Table 15.

|                  |          | DP-LDM (Ours)   | DP-DM | DP-Diffusion | DP-HP | PEARL | DPGANr |
| ---------------- | -------- | --------------- | ----- | ------------ | ----- | ----- | ------ |
| $\epsilon = 10$  | CNN      | 97.4± 0.1       | **98.1** | -         | -     | 78.8  | 95.0   |
|                  | WRN      | 97.5± 0.0       | -     | **98.6**     | -     | -     | -      |
| $\epsilon = 1$   | CNN      | **95.9± 0.1**   | 95.2  | -            | 81.5  | 78.2  | 80.1   |
|                  | # params | **0.8M**        | 1.75M | 4.2M         | -     | -     | -      |
|                  | GPU Hours | **10h**        | 192h  | -            | -     | -     | -      |

### A.2.1 CHOOSE PUBLIC DATASET FROM SVHN, KMNIST AND EMNIST FOR MNIST

We consider SVHN (street number digit but 3 channels), KMNIST (Japanese character in MNIST format 1 channel), and EMNIST (English letter in MNIST format 1 channel) as public dataset candidates. In general, if the two datasets are close to each other, then transfer learning from one to another is assumed to achieve better results within few iterations. To choose the best candidate, we compute the FID between each one with respect to MNIST and pick EMNIST as the public dataset in the end. We also did additional experiments using SVHN and KMNIST under the same privacy condition $\epsilon = 10, \delta = 10^{-5}$ in Table 8, which furthermore verifies our choice of EMNIST.

Table 8: FID scores for datasets with respect to MNIST. We also pretrained LDMs using SVHN and KMNIST then fine-tuned with MNIST, and list the best CNN accuracy here respectively.

| Dataset pair        | FID   | Best CNN accuracy |
| ------------------- | ----- | ----------------- |
| (SVHN, MNIST)       | 231.6 | 94.3              |
| (KMNIST, MNIST)     | 53.7  | 96.3              |
| **(EMNIST, MNIST)** | 27.5  | 97.4              |

### A.2.2 ABLATION EXPERIMENTS FOR MNIST

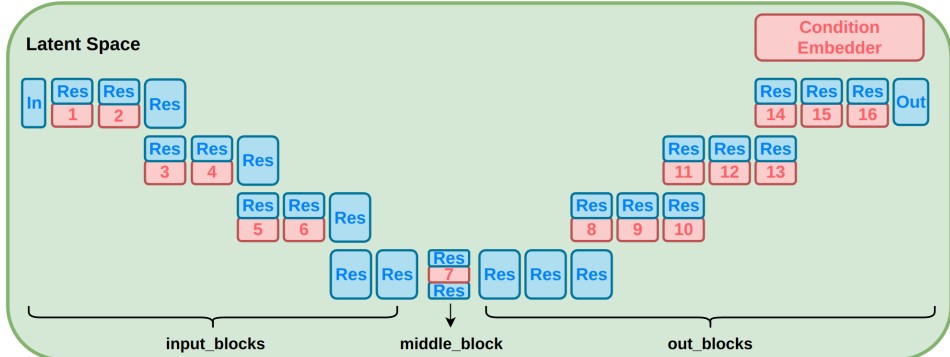

Figure 7: Unet Structure for CIFAR-10

There are 7 attention modules in the Unet structure for MNIST, 1-2 are in input_blocks, 3 is in middle_block, 4-7 are in out_blocks as illustrated in Fig. 6. Modules in blue are frozen during fine-tuning. Parameters of condition embedder is always trained. We consider fine tune only $i$-th to 7th attention modules to reduce more trainable parameters. Results for $\epsilon = 10, \delta = 10^{-5}$ are listed in Table 9. The best results is achieved when fine tune with 4-7 attention blocks, which means out_blocks are more important than others during training.

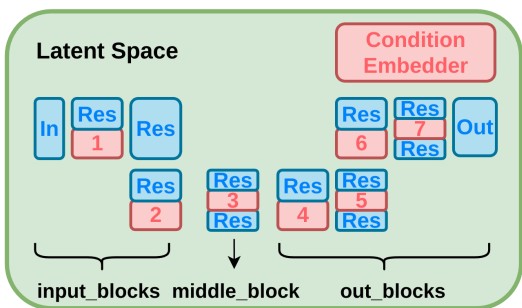

Figure 6: Unet Structure for MNIST.

Table 9: CNN accuracy and number of trainable parameters for MNIST ablation experiments with varying number of fine-tuning layers. Privacy condition is set to $\epsilon = 10, \delta = 10^{-5}$.

|  | 1-7(all) | 2-7 | 3-7 | 4-7 | 5-7 |
|---|---|---|---|---|---|
| CNN | 97.3 | 97.3 | 90 | **97.4** | 97.3 |
| # of trainable params | 1.6M | 1.5M | 1.2M | 0.8M | 0.5M |
| out of 4.6M total params | (34.3%) | (32.4%) | (25.2%) | (18.0%) | (10.9%) |

## A.3 Transferring from Imagenet to CIFAR10 distribution

Here, we provide the results for ablation experiments to test the performance of DP-LDM when fine-tuning only certain attention modules inside the pre-trained model and keeping the rest of the parameters fixed. There are 16 attention modules in total as illustrated in Fig. 7. Table 10 shows the FID obtained for $\epsilon = 1, 5, 10$ and $\delta = 10^{-5}$ for the different number of attention modules fine-tuned. The results show that fixing up to the first half of the attention layers in the LDM has a positive effect in terms of the FID (the lower the better) in our model.

We also report the different hyper-parameter settings used in ablation experiments in table Table 11.

Table 12 shows the hyper-parameters used during training ResNet9 and WRN40-4 downstream classifiers on CIFAR10 synthetic samples.

## A.4 Transferring from Imagenet to CelebA32

We also apply our model in the task of generating $32 \times 32$ CelebA images. The same pretrained autoencoder as our CIFAR-10 experiments in Section 5.1 is used, but since this experiment is for unconditional generation, we are unable to re-use the LDM. A new LDM is pretrained on Imagenet without class conditioning information, and then fine-tuned on CelebA images scaled and cropped to $32 \times 32$ resolution. Our FID results for $\delta = 10^{-6}$, $\epsilon = 1, 5, 10$ are summarized in Table 13. We

Table 10: FID scores (lower is better) for synthetic CIFAR-10 data with varying the number of fine-tuning layers and privacy guarantees. **Top row (1-16 layers):** Fine-tuning all attention modules. **Second row (5-16 layers):** Keep first 4 attention modules fixed and fine-tuning from 5 to 16 attention modules. **Third row (9-16 layers):** Keep first 8 attention modules fixed and fine-tuning from 9 to 16 attention modules. **Bottom row (13-16 layers):** Keep first 12 attention modules fixed and fine-tuning from 13 to 16 attention modules.

| DP-LDM | $\epsilon = 10$ | $\epsilon = 5$ | $\epsilon = 1$ |
|---|---|---|---|
| 1-16 layers | $25.8 \pm 0.3$ | $29.9 \pm 0.2$ | $33.0 \pm 0.3$ |
| 5 - 16 layers | $15.7 \pm 0.3$ | $21.2 \pm 0.2$ | $28.9 \pm 0.2$ |
| 9 - 16 layers | $\mathbf{8.4 \pm 0.2}$ | $\mathbf{13.4 \pm 0.4}$ | $\mathbf{22.9 \pm 0.5}$ |
| 13 - 16 layers | $12.3 \pm 0.2$ | $18.5 \pm 0.2$ | $25.2 \pm 0.5$ |

Table 11: DP-LDM hyper-parameter setting on CIFAR-10 for different ablation experiments.

|  |  | $\epsilon = 10$ | $\epsilon = 5$ | $\epsilon = 1$ |
|---|---|---|---|---|
| 1-16 layers (24.4M parameters) | batch size | 1000 | 2000 | 1000 |
|  | clipping norm | $10^{-5}$ | $10^{-5}$ | $10^{-3}$ |
|  | learning rate | $10^{-6}$ | $10^{-6}$ | $10^{-6}$ |
|  | epochs | 30 | 30 | 10 |
| 5-16 layers (20.8M parameters) | batch size | 5000 | 5000 | 2000 |
|  | clipping norm | $10^{-6}$ | $10^{-5}$ | $10^{-3}$ |
|  | learning rate | $10^{-6}$ | $10^{-6}$ | $10^{-5}$ |
|  | epochs | 50 | 50 | 10 |
| 9-16 layers (10.2M parameters) | batch size | 2000 | 2000 | 5000 |
|  | clipping norm | $10^{-6}$ | $10^{-6}$ | $10^{-2}$ |
|  | learning rate | $10^{-6}$ | $10^{-6}$ | $10^{-6}$ |
|  | epochs | 30 | 30 | 10 |
| 13-16 layers (4M parameters) | batch size | 2000 | 2000 | 2000 |
|  | clipping norm | $10^{-6}$ | $10^{-6}$ | $10^{-2}$ |
|  | learning rate | $10^{-6}$ | $10^{-6}$ | $10^{-6}$ |
|  | epochs | 30 | 30 | 10 |

achieve similar results to DP-MEPF for $\epsilon = 5$ and $\epsilon = 10$. As with our results at $64 \times 64$ resolution, our LDM model does not perform as well in higher privacy settings ($\epsilon = 1$). Sample images are provided in Figure 8

# B  HYPERPARAMETERS

Here we provide an overview of the hyperparameters of the pretrained autoencoder in Table 14, hyperparameters of the pretrained diffusion models in Table 15. Note that $base\ learning\ rate$ is the one set in the yaml files. The real learning rate passed to the optimizer is $accumulate\_grad\_batches \times num\_gpus \times batch\ size \times base\ learning\ rate$.

Table 12: Hyperparameters for downstream classification ResNet9 and WRN40-4 trained on CIFAR10 synthetic data

|  | ResNet9 | WRN40-4 |
|---|---|---|
| learning rate | 0.5 | 0.1 |
| batch size | 512 | 1000 |
| epochs | 10 | 10000 |
| optimizer | SGD | SGD |
| label smoothing | 0.1 | 0.0 |
| weight decay | $5 \cdot 10^{-4}$ | $5 \cdot 10^{-4}$ |
| momentum | 0.9 | 0.9 |

Table 13: CelebA FID scores (lower is better) for images of resolution $32 \times 32$ comparing with results from DPDM (Dockhorn et al., 2023), DP Sinkhorn (Cao et al., 2021), and DP-MEPF (Harder et al., 2023).

|  | $\epsilon = 10$ | $\epsilon = 5$ | $\epsilon = 1$ |
|---|---|---|---|
| **DP-LDM** (ours, average) | $16.2 \pm 0.2$ | $16.8 \pm 0.3$ | $25.8 \pm 0.9$ |
| **DP-LDM** (ours, best) | **16.1** | **16.6** | 24.6 |
| DP-MEPF ($\phi_1$) | 16.3 | 17.2 | **17.2** |
| DP-GAN (pre-trained) | 58.1 | 66.9 | 81.3 |
| DPDM (no public data) | 21.2 | - | 71.8 |
| DP Sinkhorn (no public data) | 189.5 | - | - |

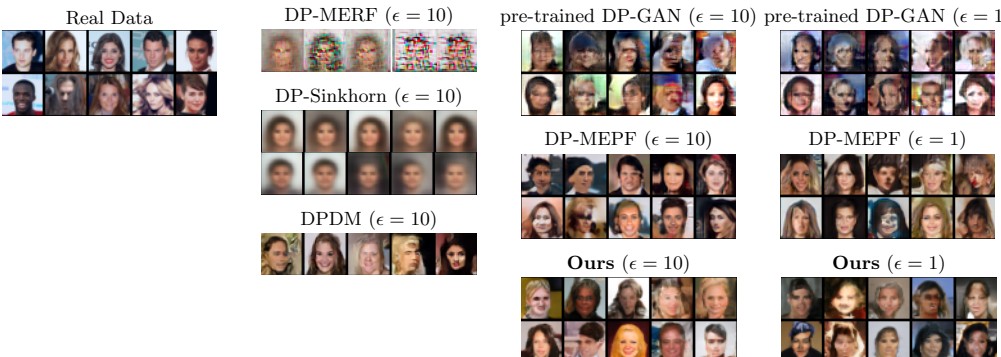

Figure 8: Synthetic $32 \times 32$ CelebA samples generated at different levels of privacy. Samples for DP-MERF and DP-Sinkhorn are taken from (Cao et al., 2021), DPDM samples are taken from (Dockhorn et al., 2023), and DP-MEPF samples are taken from (Harder et al., 2023).

Table 16 shows the hyperparameters we used for fine-tuning on MNIST. Table 17 shows the hyperparameters we used for CelebA32. Table 18 shows the hyperparameters we used for CelebA64. Table 19 shows the hyperparameters we used for text-to-image CelebAHQ generation. Table 20 shows the hyperparmeters we used for class-conditioned CelebAHQ generation.

## C  ADDITIONAL SAMPLES

Table 14: Hyperparameters for the pretrained autoencoders for different datasets.

| | EMNIST (to MNIST) | ImageNet (to CIFAR10) | ImageNet (to CelebA 32) | ImageNet (to CelebA 64) |
|---|---|---|---|---|
| Input size | 32 | 32 | 32 | 64 |
| Latent size | 4 | 16 | 16 | 32 |
| $f$ | 8 | 2 | 2 | 2 |
| $z$-shape | $4 \times 4 \times 3$ | $16 \times 16 \times 3$ | $16 \times 16 \times 3$ | $64 \times 64 \times 3$ |
| Channels | 128 | 128 | 128 | 192 |
| Channel multiplier | [1,2,3,5] | [1,2] | [1,2] | [1,2] |
| Attention resolutions | [32,16,8] | [16, 8] | [16, 8] | [16,8] |
| num_res_blocks | 2 | 2 | 2 | 2 |
| Batch size | 50 | 16 | 16 | 16 |
| Base learning rate | $4.5 \times 10^{-6}$ | $4.5 \times 10^{-6}$ | $4.5 \times 10^{-6}$ | $1.0 \times 10^{-6}$ |
| Learning rate | $4.5 \times 10^{-4}$ | $1.4 \times 10^{-4}$ | $1.4 \times 10^{-4}$ | $1.4 \times 10^{-4}$ |
| Epochs | 50 | 2 | 2 | - |
| GPU(s) | 1 NVIDIA V100 | 1 NVIDIA RTX A4000 | 1 NVIDIA RTX A4000 | 1 NVIDIA V100 |
| Time | 8 hours | 1 day | 1 day | 1 day |

Table 15: Hyperparameters for the pretrained diffusion models for different datasets.

| | EMNIST (to MNIST) | ImagNnet (to CIFAR10) | ImageNet (to CelebA 32) | ImageNet (to CelebA64) |
|---|---|---|---|---|
| input size | 32 | 32 | 32 | 64 |
| latent size | 4 | 16 | 16 | 32 |
| $f$ | 8 | 2 | 2 | 2 |
| $z$-shape | $4 \times 4 \times 3$ | $16 \times 16 \times 3$ | $16 \times 16 \times 3$ | $32 \times 32 \times 3$ |
| channels | 64 | 128 | 192 | 192 |
| channel multiplier | [1,2] | [1,2,2,4] | [1,2,4] | [1,2,4] |
| attention resolutions | [1,2] | [1,2,4] | [1,2,4] | [1,2,4] |
| num_res_blocks | 1 | 2 | 2 | 2 |
| num_heads | 2 | 8 | - | 8 |
| num_head_channels | - | - | 32 | - |
| batch size | 512 | 500 | 384 | 256 |
| base learning rate | $5 \times 10^{-6}$ | $1 \times 10^{-6}$ | $5 \times 10^{-7}$ | $1 \times 10^{-6}$ |
| learning rate | $2.6 \times 10^{-3}$ | $5 \times 10^{-4}$ | $2 \times 10^{-4}$ | $2.6 \times 10^{-4}$ |
| epochs | 120 | 30 | 13 | 40 |
| # trainable parameters | 4.6M | 90.8M | 162.3M | 72.2M |
| GPU(s) | 1 NVIDIA V100 | 1 NVIDIA RTX A4000 | 1 NVIDIA V100 | 1 NVIDIA V100 |
| time | 6 hours | 7 days | 30 hours | 10 days |
| use_spatial_transformer | True | True | False | False |
| cond_stage_key | class_label | class_label | - | - |
| conditioning_key | crossattn | crossattn | - | - |
| num_classes | 26 | 1000 | - | - |
| embedding dimension | 5 | 512 | - | - |
| transformer depth | 1 | 1 | - | - |

DP-LDM (Ours)                                    DP-MEPF

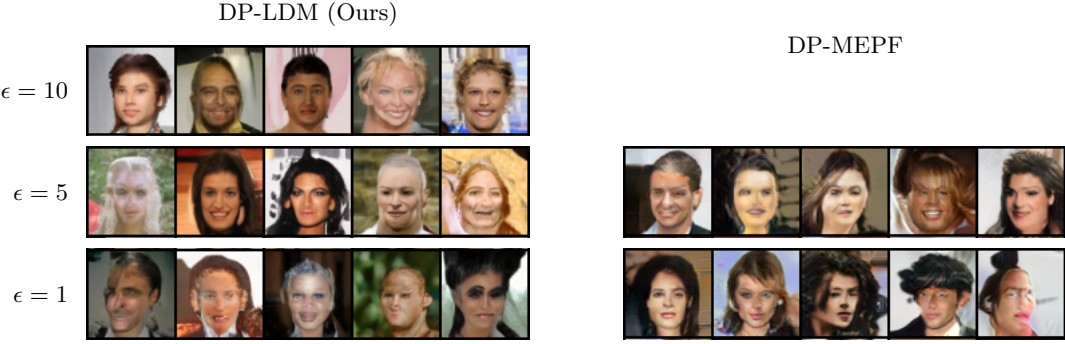

Figure 9: Synthetic $64 \times 64$ CelebA samples generated at different levels of privacy. Samples for DP-MEPF are taken from Harder et al. (2023).

Table 16: Hyperparameters for fine-tuning diffusion models with DP constraints $\epsilon = 10, 1$ and $\delta = 10^{-5}$ on MNIST. The "ablation" hyperparameter determines which attention modules are fine-tuned, where a value of $i$ means that the first $i - 1$ attention modules are frozen and others are trained. Setting "ablation" to $-1$ (default) fine-tunes all attention modules.

|  | $\epsilon = 10$ | $\epsilon = 1$ |
|---|---|---|
| batch size | 2000 | 2000 |
| base learning rate | $5 \times 10^{-7}$ | $6 \times 10^{-7}$ |
| learning rate | $1 \times 10^{-3}$ | $1.2 \times 10^{-3}$ |
| epochs | 200 | 200 |
| clipping norm | 0.01 | 0.001 |
| noise scale | 1.47 | 9.78 |
| ablation | 4 | -1 |
| num of params | 0.8M | 1.6M |
| use_spatial_transformer | True | True |
| cond_stage_key | class_label | class_label |
| conditioning_key | crossattn | crossattn |
| num_classes | 26 | 26 |
| embedding dimension | 13 | 13 |
| transformer depth | 1 | 1 |
| train_condition_only | True | True |
| attention_flag | spatial | spatial |
| # condition params | 338 | 338 |

Table 17: Hyperparameters for fine-tuning diffusion models with DP constraints $\epsilon = 10, 5, 1$ and $\delta = 10^{-6}$ on CelebA32.

|  | $\epsilon = 10$ | $\epsilon = 5$ | $\epsilon = 1$ |
|---|---|---|---|
| batch size | 8192 | 8192 | 2048 |
| base learning rate | $5 \times 10^{-7}$ | $5 \times 10^{-7}$ | $5 \times 10^{-7}$ |
| learning rate | $4 \times 10^{-3}$ | $4 \times 10^{-3}$ | $1 \times 10^{-3}$ |
| epochs | 20 | 20 | 20 |
| clipping norm | $5.0 \times 10^{-4}$ | $5.0 \times 10^{-4}$ | $5.0 \times 10^{-4}$ |
| ablation | -1 | -1 | -1 |
| use_spatial_transformer | False | False | False |
| cond_stage_key | - | - | - |
| conditioning_key | - | - | - |
| num_classes | - | - | - |
| embedding dimension | - | - | - |
| transformer depth | - | - | - |
| train_attention_only | True | True | True |

Table 18: Hyperparameters for fine-tuning diffusion models with DP constraints $\epsilon = 10, 5, 1$ and $\delta = 10^{-6}$ on CelebA64.

|  | $\epsilon = 10$ | $\epsilon = 5$ | $\epsilon = 1$ |
|---|---|---|---|
| batch size | 8192 | 8192 | 8192 |
| base learning rate | $1 \times 10^{-7}$ | $1 \times 10^{-7}$ | $1 \times 10^{-7}$ |
| learning rate | $8.2 \times 10^{-4}$ | $8.2 \times 10^{-4}$ | $8.2 \times 10^{-4}$ |
| epochs | 70 | 70 | 70 |
| clipping norm | $5.0 \times 10^{-4}$ | $5.0 \times 10^{-4}$ | $5.0 \times 10^{-4}$ |
| ablation | -1 | -1 | -1 |
| use_spatial_transformer | False | False | False |
| cond_stage_key | - | - | - |
| conditioning_key | - | - | - |
| num_classes | - | - | - |
| embedding dimension | - | - | - |
| transformer depth | - | - | - |
| train_attention_only | True | True | True |

Table 19: Hyperparameters for fine-tuning diffusion models with DP constraints $\epsilon = 10, 1$ and $\delta = 10^{-5}$ on text-conditioned CelebAHQ.

|  | $\epsilon = 10$ | $\epsilon = 1$ |
|---|---|---|
| batch size | 256 | 256 |
| base learning rate | $1 \times 10^{-7}$ | $1 \times 10^{-7}$ |
| learning rate | $2.6 \times 10^{-5}$ | $2.6 \times 10^{-5}$ |
| epochs | 10 | 10 |
| clipping norm | 0.01 | 0.01 |
| noise scale | 0.55 | 1.46 |
| ablation | -1 | -1 |
| num of params | 280M | 280M |
| use_spatial_transformer | True | True |
| cond_stage_key | caption | caption |
| context_dim | 1280 | 1280 |
| conditioning_key | crossattn | crossattn |
| transformer depth | 1 | 1 |

Table 20: Hyperparameters for fine-tuning diffusion models with DP constraints $\epsilon = 10, 5, 1$ and $\delta = 10^{-5}$ on class-conditional CelebAHQ.

|  | $\epsilon = 10$ | $\epsilon = 5$ | $\epsilon = 1$ |
|---|---|---|---|
| batch size | 2048 | 2048 | 2048 |
| base learning rate | $1 \times 10^{-7}$ | $1 \times 10^{-7}$ | $1 \times 10^{-7}$ |
| learning rate | $2.0 \times 10^{-4}$ | $2.0 \times 10^{-4}$ | $2.0 \times 10^{-4}$ |
| epochs | 50 | 50 | 50 |
| clipping norm | $5.0 \times 10^{-4}$ | $5.0 \times 10^{-4}$ | $5.0 \times 10^{-4}$ |
| ablation | -1 | -1 | -1 |
| use_spatial_transformer | True | True | True |
| cond_stage_key | class_label | class_label | class_label |
| context_dim | 512 | 512 | 512 |
| conditioning_key | crossattn | crossattn | crossattn |
| transformer depth | 1 | 1 | 1 |

