# OpenReview forum: "Differentially Private Latent Diffusion Models"
_ICLR.cc/2024/Conference — Submitted to ICLR 2024_

### Official Review · Reviewer_2m5o · 2023-10-27

**Soundness:** 3 good
**Presentation:** 3 good
**Contribution:** 2 fair
**Rating:** 5
**Confidence:** 4

**Summary:**

This paper studies applying fine-tuning techniques and public data in DP-SGD to privately train diffusion model. Resorting to public data for pre-train and applying DP-SGD to only fine-tune a small fraction of model parameters, the authors show the improvement both from training time and the performance.

**Strengths:**

This paper is well-written and all the ideas are clearly presented. As an empirical paper, the authors detail the selections of hyper-parameters and model architecture.

**Weaknesses:**

Though the experiments are comprehensive and solid, the key ideas of this paper are relatively simple. The curse of dimensionality of DP is already a well-known problem, and for implementation of DP-SGD, especially in supervised learning tasks, pretrain with public data and fine-tuning have been extensively studied. It is interesting to see the application of those techniques in privately training diffusion model, but I am afraid that the new insights provided by this paper are not enough. For example, with the assistance of public data, one may also consider further improvement such as gradient embedding or low-rank adaptation in [1,2].

In addition, since the authors propose fine-tuning only a small fraction of parameters, the produced efficiency improvement with a smaller noise required are clearly-expected consequences. I also note that fine-tuning may also come with a tradeoff. For example in Table 7, in low privacy regime, DP-DM with full training can out-performance proposed methods. But such tradeoff seems not being fully studied.


Minor issue: The main document should only contain a 9-page main body and references, while the authors also attach the appendix.

[1] Differentially private fine-tuning of language models
[2] Do not Let Privacy Overbill Utility: Gradient Embedding Perturbation for Private Learning

**Questions:**

I have several suggestions for the authors to further improve this paper. At a high-level, as an empirical paper, I would suggest a more comprehensive study on the influence of network architecture selection on DP-SGD; Also, to refine the fine-tuning results, the authors can also through searching to determine the fraction or layers of the diffusion model to fine-tune, as some general instruction for future work to design the fine-tuning strategy; Finally, given that public data is assumed, the author may consider how to fully exploit to refine the DP-SGD with less noise using embedding or low rank approximation.

---

> ### Author Response · Authors · 2023-11-18
>
> We would like to thank the reviewer for the feedback and suggestions for further improvement. We will add them to our final version of paper.
> - About Novelty:
> We would like to reiterate the main contributions of this work. Diffusion models are currently the state-of-the-art method in image generation. There are currently two published methods that attempt to make diffusion models differentially private. DP-DM performs well on simple datasets such as MNIST and FashionMNIST, but does not perform well on more complex datasets such as CIFAR10 and CelebA32 and 64. This method also requires a long training time as a diffusion model is trained from scratch. DP-Diffusion attempts to overcome this issue by pretraining a diffusion model on public data and then fine-tuning on private data. However, this fine-tuning is done on all of the parameters of the diffusion model and is still computationally costly. Our method incorporates the benefits of all these attempts. In particular, we also use diffusion models, as DMs model the complex image distributions very well. We also use public data to reduce the training time (which leads to a smaller privacy loss). On top of these ingredients, we incorporate the latent diffusion models that reduce the dimension of diffusion process and selectively fine-tune attention modules in the LDMs to further reduce the fine-tuning time and dimensionality of the parameters to be fine-tuned. The resulting algorithm achieves state-of-the-art results in image generation with differential privacy. There is no other work that achieves this level of accuracy using diffusion models. Thanks to our particular model choice, we can furthermore generate realistic images at the scale of 256x256 and text-to-image generation with differential privacy, which was not explored before.
> - Comprehensive study on the influence of network architecture selection on DP-SGD: We have these results in our appendix **already**, where we tested a different number of layers for  LDM, different downsampling ratio (f) and different number of channels for auto-encoders, and ablation studies given in several sections in our Appendix. These are all standard things to search when using LDM, which we diligently delivered already. If there are any other things we should consider doing, please let us know.
> - General instruction for fine-tuning strategy: For this, we illustrated the accuracy versus which layer’s attention modules to fine-tune, and also how to pick a good public data given a private dataset. Overall, we found that fine-tuning the layers in Out Blocks were more useful than those in Input Blocks, at a fixed privacy budget. This outcome was consistent through all datasets we tested. So this is one strategy we will suggest for future use of this work in a new task. We can use FID (or a privatized FID, see our comment to c8Le) as a proxy to judge if a particular candidate public data is more useful than others, which is another strategy for users. If there are other things the reviewer thinks we should add other than these strategies, please let us know.
> - Given that public data is assumed, the author may consider how to fully exploit to refine the DP-SGD with less noise: In our method, we pre-train an auto-encoder (VQ-GAN based), which provides the embedding of high-dimensional inputs and creates the diffusion process in this latent space to reduce the amount of noise to add in DP-SGD, compared to running DP-SGD on the high-dimensional input space. One can additionally use gradient embedding as the reviewer suggests to further reduce the noise in DP-SGD. In fact, any of the numerous papers that aim to improve the performance of DP-SGD (moving average, data augmentation, sparsifying gradients, DP-ADAM, etc) can all be applied to our work to further improve the performance of DP-SGD.

---

> ### Author Response · Authors · 2023-11-18
>
> - Low rank approximation: We thank the reviewer for this suggestion. Our current submission studies the trade-off between the final accuracy based on which layer’s attention modules to be fine-tuned, and the privacy guarantee we provide. Once we include LoRA or other parameter-efficient LLM training schemes, this fore-mentioned trade-off takes an extra dimension. Namely, once we approximate the parameters by a few low-rank matrices, we reduce the dimension of the parameters to fine-tune with DP-SGD (which is a good thing). This, however, introduces a loss of information about the parameters (which is a bad thing). As a result, now we need to study the trade-off among the three: (a) which layer’s attention modules to fine-tune (so far fine-tuning the layers in out block is preferred) ; (b) how accurate the approximation gets (which prefers a higher rank); and  (c) how the resulting model performs in the presence of noise due to DP-SGD (which prefers a lower rank). This trade-off might exacerbate when there is a larger gap between the public and private datasets we consider. We are currently looking into the possibility of plugging in the LoRA package into our code base to test this trade-off. **Please check our updated results at** https://openreview.net/forum?id=FLOxzCa6DS&noteId=7Huv2tefxz
>
> - Table 7: As the reviewer observed, DPDM outperformed our method when tested on MNIST. This result, however, is not transferable to other more complex datasets, where pre-training a larger model (LDM) and fine-tuning a small number of selected parameters made a larger accuracy increase than training the entire DM with DP-SGD (see Table 13, CelebA32). Here is some more comparison to DPDM, where the numbers we took from their Appendix (https://openreview.net/pdf?id=pX21pH4CsNB) for CelebA64 and CIFAR10:
> | FID                           | **DP-LDM(ours)** | DPDM |
> |-------------------------------|------------------|------|
> | CelebA64 (eps=10, delta=1e-6) | **14.2**         | 78.3 |
> | CIFAR10 (eps=10, delta=1e-5)  | **8.4**          | 97.7 |
>
> - Minor issue: The main document should only contain a 9-page main body and references, while the authors also attach the appendix. Thanks for brining this up. We double checked the author guideline at https://iclr.cc/Conferences/2024/AuthorGuide, it is said that either way is allowed.

---

> > ### Author Response · Authors · 2023-11-23
> >
> > As the rebuttal period is ending soon, we would like to ask the reviewer whether we have resolved your concerns? If the reviewer has more questions to ask or needs clarifications, we are happy to discuss.

---

> > > ### Comment · Reviewer_2m5o · 2023-11-23
> > >
> > > Thanks for your response! I think the experimental part is clear but I am still a bit worried about the technical novelty and would like to keep my score.

---

### Official Review · Reviewer_muGo · 2023-10-28

**Soundness:** 3 good
**Presentation:** 3 good
**Contribution:** 2 fair
**Rating:** 5
**Confidence:** 4

**Summary:**

The paper introduces a privacy-preserving diffusion model achieved by fine-tuning the attention layer of the latent-diffusion model. The authors pre-trained the model using publicly available data to avoid consuming the privacy budget.

**Strengths:**

The paper demonstrates state-of-the-art results in image generation by allowing pre-training with publicly available data.

**Weaknesses:**

The most significant thing lack of novelty: The paper simply fine-tune the pre-trained public model with a similar technique in You & Zhao (2023). This is unremarkable because reducing the fine-tuning space is mentioned in You & Zhao (2023).

Here are minor weaknesses:
The notations $\Delta$ and $\nabla$ used for the encoder and decoder might be confusing and could be clarified. There are some typographical errors, such as "$B$" instead of "$B_p$" in Algorithm 1. Furthermore, in Table 2, it's unclear how the privacy budget of synthetic data was handled when reporting 88.3% accuracy.
For the choice of public data, due to the unavailability of private data, you should not calculate FID between private and public data. Since EMNIST for MNIST dataset and ImageNet for other dataset are commonly used for public dataset, using them makes sense. However, calculating FID to choose public data may give readers the impression that private data was accessed.

**Questions:**

Correct me if I am wrong. From my understanding, the conditioning embedder appears to function effectively only for language prompts and not for class conditions.
Can you explain how the conditioning embedder works when it has been pre-trained with public data with different labels than the private data? Does the model treat it as a class embedder, or does it treat it as random initialization?

---

> ### Author Response · Authors · 2023-11-18
>
> We thank the reviewer for the feedback. Here are the answers of each point :
>
> - Novelty: We clearly mentioned in our introduction that the use of DP-SGD to fine-tune a pre-trained model itself could be viewed unremarkable as a method. However, please pay attention to what impact this seemingly ordinary method can bring. In fact, the potential impact is well appreciated and described by other reviewers: Reviewer tmmk wrote “The paper reads incremental, but it does a decent job in finding the correct hammer, which does lead to promising results.”, and reviewer c8Le wrote “Although the proposed approach lacks novelty, DP latent diffusion models could be of great interest to the community and the results look promising. “.
> Notation on encoder and decoder: we’ve used E and D for referring to other things in the paper, so we settled with a non-standard notation for encoder/decoder. We can certainly change these to others. We’re open to suggestions, if the reviewer wants to suggest a better notation.
>
> - "encoder “ Δ and decoder ∇ ": non-standard notations: We understand that E and D are commonly used for encoders and decoders. Unfortunately, we have used D to note dataset to define differential privacy. We are considering changing encoder Δ and decoder ∇ to be Enc and Dec as used in [1,2,3]. But if the reviewer has a suggestion on which notation we should use for encoders and decoders, we’re open to it.
>
> - Table 2: There is **NO synthetic data** involved in the 88.3% accuracy. Showing this accuracy (evaluated on real test data) is quite standard in DP data generation literature, as it is used as a top accuracy synthetic data could achieve given a dataset (implying that no synthetic data can be better than real data).
>
> - For FID using private data: please see https://openreview.net/forum?id=FLOxzCa6DS&noteId=BmOgRtVX6z.
>
> - Conditioning embedder: Conditioning embedder refers to both **class** conditioning and other types of conditioning such as conditioning on prompts, blurred variants of input image, semantic maps, layouts, etc. For more information about conditioning in LDM, refer to [4]. When there’s a difference between the classes in private and public datasets, e.g., in imagenet32 with 1000 classes and cifar10 with 10 classes, we freeze the embedder corresponding to 9900 classes, while fine-tuning the pre-trained embedder for the first 10 classes. Then, we use only these fine-tuned components when generating class-conditioned images for CIFAR10.
>
> [1] ABL Larsen et al, Autoencoding beyond pixels using a learned similarity metric, PMLR, 2015
>
> [2] Liu, Yihong et al, On the Copying Problem of Unsupervised NMT: A Training Schedule with a Language Discriminator Loss, ACL, 2023
>
> [3] Sprinks, Graham, and Moens, Marie-Francine, Generating Continuous Representations of Medical Texts, NAACL, 2018
>
> [4] Rombach et al, High-Resolution Image Synthesis with Latent Diffusion Models,  CVPR, 2022

---

> > ### Author Response · Authors · 2023-11-23
> >
> > As the rebuttal period is ending soon, we would like to ask the reviewer whether we have resolved your concerns? If the reviewer has more questions to ask or needs clarifications, we are happy to discuss.

---

> > ### Comment · Reviewer_muGo · 2023-11-23
> >
> > Thank you for answering each question.
> >
> > Minor concern: if you calculate the FID using DP-EM, then the privacy budget becomes larger.  You may need to do more precise experiments for your final paper.
> >
> > Question: Can fine-tuning the model with other images, for example, camelyon17 data for cifar10, be used?
> >  In your experiment, EMNIST is very close to MNIST (FID=2 for epsilon=2). Therefore, simply generating EMNIST images (eliminating privacy concerns) may outperform this result.

---

> > > ### Author Response · Authors · 2023-11-23
> > >
> > > Sorry, we do not understand the purpose of pre-training the model with Camelyon17 then fine-tuning the model for Cifar10. For transfer learning to work well, the expectation is that the public (source) and private (target) data distributions have a descent amount of overlapping support. The cases we consider in our paper are all commonly used private/public data pairs: Imagenet -> Cifar10, Imagenet -> CelebA, Digit data (like SVHN, EMNIST, KMNIST)->MNIST. If one insists on using completely different dataset as a public (source) dataset, the performance of any transfer learning framework would boil down to the performance of training from scratch.

---

> > > > ### Comment · Reviewer_muGo · 2023-11-23
> > > >
> > > > I still think this paper significantly lacks novelty.
> > > > I think additional analysis should be contained in your final paper.
> > > > However, I will increase my score looking forward to adding an ablation study compared with DPDM, which does not use public data but gets a similar result in large epsilon.

---

> > > > > ### Author Response · Authors · 2023-11-23
> > > > >
> > > > > Thanks for increasing the score.
> > > > >
> > > > > Regarding comparison with DPDM, which focused on DP generation on simpler dataset such as MNIST, we did the comparison in section 5.3 and the results are illustrated in appendix due to space limit. As shown in Table 7, We achieved higher CNN accuracy with small epsilon, and DPDM outperforms us on larger epsilon. However, Table 7 also showed the computational resources spent, and our method takes much fewer GPU hours, which is also the aim of our paper -- to greatly reduce the trainable parameters and the training time.
> > > > >
> > > > > We see that DPDM outperformed our method when tested on MNIST with large epsilon. This result, however, is not transferable to other more complex datasets, where pre-training a larger model (LDM) and fine-tuning a small number of selected parameters made a larger accuracy increase than training the entire DM with DP-SGD (see Table 13, CelebA32). Here is some more comparison to DPDM, where the numbers we took from their Appendix (https://openreview.net/pdf?id=pX21pH4CsNB) for CelebA64 and CIFAR10:
> > > > >
> > > > > | FID                           | **DP-LDM(ours)** | DPDM |
> > > > > |-------------------------------|------------------|------|
> > > > > | CelebA64 (eps=10, delta=1e-6) | **14.2**         | 78.3 |
> > > > > | CIFAR10 (eps=10, delta=1e-5)  | **8.4**          | 97.7 |

---

### Official Review · Reviewer_c8Le · 2023-11-01

**Soundness:** 3 good
**Presentation:** 3 good
**Contribution:** 2 fair
**Rating:** 6
**Confidence:** 4

**Summary:**

The paper proposes to train differentially private latent diffusion models for generating DP synthetic images. Compared to training diffusion diffusions on the image spaces, training latent diffusion models reduces the number of parameters and therefore could be more friendly (in terms of computational cost and privacy-utility trade-off) in DP settings. To further reduce the number of training parameters, the paper proposes to only fine-tune the attention layers and the condition embedders. Experiments show that the proposed method achieves state-of-the-art privacy-utility trade-offs on several benchmark datasets.

**Strengths:**

* The paper is well-written.
* Given the widespread and successful use of latent diffusion models in non-DP settings, exploring whether they can help DP synthetic data generation is very important and timely. This paper demonstrates a practical pathway for doing it. The open-source code could be very useful to the community.
* The results look promising.

**Weaknesses:**

* The proposed approach is a straightforward application of existing techniques.

**Questions:**

Although the proposed approach lacks novelty, DP latent diffusion models could be of great interest to the community and the results look promising. Therefore, I am leaning toward a positive score. However, it would be great if the authors could clarify the following questions and I will adjust the score accordingly.

* The paper proposes to fine-tune only the attention layers and the condition embedders to reduce the number of fine-tuning parameters. One of the most commonly used approaches in both DP and non-DP communities is to do LoRA, adapter, or compacter fine-tuning (see [3] for an example). It would be better to comment on or experimentally compare with such approaches.

* Table 7 in the appendix shows the GPU hours for training DP-LDM and DP-DM. Could you clarify if that includes the pre-training cost for both methods? If yes, it would be clearer to break down the time into pre-training and fine-tuning stages. If not, it would be better to include pre-training costs as well, as at least in the experiments of this paper, customized pre-training for each dataset has to be done.

* Introduction claims that "DPSGD ... does not scale well for large models that are necessary for learning complex distributions." It is not necessarily true. Prior work has demonstrated that DP-SGD works well with large language models. See [1,2] for some examples.

* What does "average" and "best" mean in Table 4?

* Section 5.3 discusses the process of selecting pre-training datasets. However, this selection process needs to use private data and therefore is NOT DP. Please refer to [4] for an example of how to select pre-training **dataset** in a DP fashion, and [5] for an example of how to select pre-training **samples** in a DP fashion. According to the results in the prior work, I guess that the selection between SVHN, KMNIST, and MNIST would only incur a small privacy cost. Still, the paper should at least discuss this issue (i.e., the privacy cost of this dataset selection step is ignored) and the related work, if not redoing the experiments.

* Table 8 in the appendix: what does "Best" mean in "Best CNN accuracy"?

* Table 10 shows that the results are sensitive to the selection of fine-tuning layers, especially in regimes with a high privacy budgets. It would be better to discuss hypotheses about why fine-tuning 9-16 layers is the best and provide recommendations for practitioners on how to choose this hyper-parameter for new datasets.

* The line after Eq. 1: x_t is not defined.

* Section 2.2 states that "A single entry difference could come from either replacing or removing one entry from the dataset D." While both definitions (replacing vs. removing) are valid and used in practice, they result in different DP bounds as the sensitivity is different. The paper should be clear which definition is used in the experiments.

* The paragraph after Eq. 3: in the definition of K and V, should \phi(x) be \phi(y)?

* Step 4 in algorithm 1: N(0, \sigma^2C^2I) should be  1/B N(0, \sigma^2C^2I)

* Related work: a space is missing in "(GANS)(Goodfellow et al.,"

* Related work: "Lin et al. (2023) do privatize" should be "Lin et al. (2023) do **not** privatize"

* Section 5: "complexity : the" should be "complexity: the"

[1] Li, Xuechen, et al. "When Does Differentially Private Learning Not Suffer in High Dimensions?." Advances in Neural Information Processing Systems 35 (2022): 28616-28630.

[2] Anil, Rohan, et al. "Large-scale differentially private BERT." arXiv preprint arXiv:2108.01624 (2021).

[3] Yu, Da, et al. "Differentially private fine-tuning of language models." arXiv preprint arXiv:2110.06500 (2021).

[4] Hou, Charlie, et al. "Privately Customizing Prefinetuning to Better Match User Data in Federated Learning." arXiv preprint arXiv:2302.09042 (2023).

[5] Yu, Da, et al. "Selective Pre-training for Private Fine-tuning." arXiv preprint arXiv:2305.13865 (2023).

---

> ### Author Response · Authors · 2023-11-18
>
> We thank the reviewer for the positive feedback and detailed suggestion. Thanks for pointing out the typos and the parts that need to be clarified. We have fixed these and will add further improvement according to the review's feedback to our final version.
>
> Below are the answers to each point:
>
> - Using LoRA in our model: Thanks for the suggestion. Our current submission studies the trade-off between the final accuracy based on which layer’s attention modules to be fine-tuned, and the privacy guarantee we provide. Once we include LoRA or other parameter-efficient LLM training schemes, this fore-mentioned trade-off takes an extra dimension. Namely, once we approximate the parameters by a few low-rank matrices, we reduce the dimension of the parameters to fine-tune with DP-SGD (which is a good thing). This, however, introduces a loss of information about the parameters (which is a bad thing). As a result, now we need to study the trade-off among the three: (a) which layer’s attention modules to fine-tune (so far fine-tuning the layers in out block is preferred) ; (b) how accurate the approximation gets (which prefers a higher rank); and  (c) how the resulting model performs in the presence of noise due to DP-SGD (which prefers a lower rank). This trade-off might exacerbate when there is a larger gap between the public and private datasets we consider. We are currently looking into the possibility of plugging in the LoRA package into our code base to test this trade-off. We will update this rebuttal once we get the results.
>
> - Clarification for the GPU hours spent: Sorry for the confusion. Table 7 does not contain the pretraining hours. There are three steps in our method, first pretraining an autoencoder takes 8 hours as listed in Table 14, second pretraining the LDM takes 6 hours as listed in Table 15, third fine-tuning the LDM to make it DP takes 10 hours, which is listed in Table 7. The DP-DM method does not contain any pretraining, they trained from scratch, which takes 192 hours in total.
>
> - Privacy definition: Yes, we use the inclusion/exclusion definition of DP, as this is what Opacus uses. We will make this clear in our paper.
>
> - DPSGD scaling issues: We’re aware of [1,2] that the reviewer cited. The finding of these papers is that when the gradients can be characterized by a few principal components (this happens in the case of fine-tuning LLMs), the scaling issue of DP-SGD is lifted. However, in general, this issue is present in most settings of training deep neural network models.
>
> - For Table 4, best result is the highest we achieved, avg result is the avg we have by taking three runs with different seeds.
>
> - Table 8, we did the hyperparmeter search for each dataset, "Best CNN accuracy" is the highest score we achieved for individual dataset, which shows out choice of EMNIST will achieve best performance among those three.
>
> - Why fine-tuning 9-16 layers is the best and provide recommendations : Our results suggest that empirically finetuning attention modules in out_blocks outperform finetuning those input_blocks. This is consistent with the results of [6]. We sould recommend this strategy to users if their privacy budget is limited. We will add this discussion to our final version.
>
>
> [6] You, Fuming and Zhao, Zhou, Transferring Pretrained Diffusion Probabilistic Models, https://openreview.net/forum?id=8u9eXwu5GAb, 2023

---

> ### Author Response · Authors · 2023-11-18
>
> - FID using private data: Whether it’s necessary to add noise to FID depends on whether we release these intermediate FID scores for choosing the public dataset. In our submission, we did not consider privatizing FID, as we had no intention of releasing these FID scores. One might argue that even if the intermediate FID scores are not released, the selection itself is already embedded in the pre-trained model, which is then used for private fine-tuning. So, to be thorough, we can consider using a DP mechanism to privatize the selection step. Recall that FID is calculated by two statistics, mean and covariance of features from a pre-trained inception network, so using a Gaussian mechanism to privatize the mean and covariance of the private data before computing FID can be used [7]. A similar idea was used in DP-MEPF[8], which uses the mean feature difference between private and public data to determine early stopping. Considering the privatized mean difference only makes sense, as the noise scale for mean perturbation is O(d), while that for covariance perturbation is O(d^2).  As shown below, the privatized mean alone, and both mean and covariance do not alter the rank of the final FID scores significantly. This stems from the fact that the sensitivity of the mean and covariance is extremely small for datasets we consider, i.e., on the order of 1 over the number of training data. So privatizing or not privatizing FID does not change the conclusion about which dataset is more useful to use as public data for pre-training. We will include this point in our final version.
>
> Here is the FID when we privatize both mean and covariance, the eps is the total value of eps for mean and eps for covariance, basically we set eps for mean = eps for covariance.
> | total eps  | 0.1         | 0.5        | 1          | 2          |
> |------------|-------------|------------|------------|------------|
> | SVHN       | 66.7628     | 14.7779    | 7.6202     | 4.1        |
> | KMNIST     | 80.6722     | 17.2918    | 8.8239     | 4.5922     |
> | **EMNIST** | **37.5071** | **7.9408** | **4.1476** | **2.1157** |
>
> Here is the mean difference when we privatize mean only:
> | eps        | 0.1        | 0.5        | 1          | 2          |
> |------------|------------|------------|------------|------------|
> | SVHN       | 0.2698     | 0.2668     | 0.2666     | 0.2665     |
> | KMNIST     | 0.0662     | 0.064      | 0.0639     | 0.064      |
> | **EMNIST** | **0.0204** | **0.0198** | **0.0197** | **0.0197** |
>
> [7] Park, Mijung et al, DP-EM: Differentially Private Expectation Maximization, AISTATS, 2017
>
> [8] Harder, Frederik et al, Pre-trained perceptual features improve differentially private image generation, TMLR, 2023

---

> > ### Comment · Reviewer_c8Le · 2023-11-21
> > **Thank you**
> >
> > Thank the authors for the reply! ICLR allows revisions so the authors do not have to wait till the final version to update the paper. Looking forward to seeing the LoRA results, which will strengthen the paper. I will keep the score for now.

---

> ### Author Response · Authors · 2023-11-22
> **LoRA Results**
>
> As requested, we performed additional experiments after incorporating LoRA into our model. Our results for unconditional CelebA generation at 64x64 resolution (trained for 50 epochs) are summarized in the following table of FID values. Each FID value is computed as the average of three runs. The final column in the table contains results using DP-LDM for comparison, though we note that the FIDs are slightly higher (worse) than in tables 4 and 5 in our main paper because these models were trained for 50 epochs instead of 70.
>
> |  |  |  | LoRA |  |  | DP-LDM |
> |---|---|---|---|---|---|---|
> |  |  |  | Rank |  |  |  |
> |  | 1 | 2 | 4 | 8 | 64 |  |
> | $\epsilon=1$ | 25.8 | 25.0 | 23.8 | 23.0 | 25.5 | 21.7 |
> | $\epsilon=5$ | 22.6 | 22.0 | 20.4 | 18.3 | 18.9 | 16.2 |
> | $\epsilon=10$ | 22.0 | 19.9 | 19.0 | 18.1 | 17.4 | 15.2 |
> | # Parameters (Trainable/Total) | 20k/72M (0.03%) |  40k/72M (0.06%) | 80k/72M (0.11%) | 16k/72M (0.22%) | 1.3M/73M (1.74%) | 8.0M/72M (11.03%) |
>
> As shown in the table, when we allow for small privacy loss (epsilon=1), LoRA achieves the best results at rank 8. As we allow for larger privacy loss (epsilon=10), the accuracy of LoRA at higher rank (64) performs better. However, in all cases, the performance with LoRA is still lower than that with the full parameters fine-tuned in DP-LDM.
>
> This phenomenon is also observed in fine-tuning large language models. As a concurrent work with [2], Li et al [1] showed parameter-efficient adaptation methods (which reduce the dimensionality of updates) do not necessarily outperform a baseline method that fine-tunes all model parameters. The large batch size improves the signal-to-noise ratio (shown in Fig 3 [1]), which significantly helps improve the performance of the full-parameter fine-tuned model. We find that this is analogous to our results. When we fine-tune the parameters in the attention modules at their original ranks, we lose less information. Due to the large batch size we use (bs=8192), we maintain a relatively good signal-to-noise ratio in the gradients perturbed by DP-SGD, which leads to better performance than LoRA at the same privacy level.
>
> It is an intriguing question, whether the intrinsic dimensionality of the gradients are truly low, when fine-tuning DP-LDMs. Perhaps we could use the metric used in [3] to empirically evaluate the intrinsic dimension. This could be a direction to pursue for future work. We thank the reviewer for raising this question.
>
> [1]  LARGE LANGUAGE MODELS CAN BE STRONG DIFFERENTIALLY PRIVATE LEARNERS, X. Li, F. Tramer, P. Liang, T. Hashimoto, ICLR 2022.
>
> [2] Differentially Private Fine-tuning of Language Models, D. Yu, S. Naik, A. Backurs, S. Gopi, H. Inan, G. Kamath, J.Kulkarni, Y. T. Lee, A. Manoel, L. Wutschitz, S. Yekhanin, and H. Zhang., ICLR 2022.
>
> [3] Intrinsic Dimensionality Explains the Effectiveness of Language Model Fine-Tuning Aghajanyan et al., ACL-IJCNLP 2021.

---

> > ### Comment · Reviewer_c8Le · 2023-11-23
> > **Thank you**
> >
> > Thank the authors for the LoRA results! It would also be great to list the computation cost of each method. Is the computation cost of LoRA smaller than your proposed approach? Such information is useful for practitioners to choose among the options based on their budget. Since the discussion phase is closing soon, I do not expect the authors to show the results immediately. But it would be great to include all these results in the final revision.
> >
> > Given the results, I would vote for acceptance of the paper. But given the limited novelty, a score of 6 might be better suited than 8, so I will keep the score in the system.
> >
> > Thank the authors again for the experiments!

---

> > > ### Author Response · Authors · 2023-11-23
> > >
> > > Thanks for the feedback! We will include the suggested improvements in our final version!

---

> ### Author Response · Authors · 2023-11-23
> **LoRA Computational Costs**
>
> Training our LDM with LoRA for 50 epochs took 36 hours on a single NVIDIA V100. This was the same across all epsilon levels and ranks ($r \in \{1, 2, 4, 8, 64\}$). For comparison, our original LDM training completed 50 epochs in 32 hours on a single NVIDIA V100. Note that after incorporating LoRA into our model, the training time **increased**. We believe this is because LoRA introduces new weight matrices into the model, increasing the amount of computation required in the backward pass. Despite fewer tunable parameters in the model (see the updated table in https://openreview.net/forum?id=FLOxzCa6DS&noteId=7Huv2tefxz), the same gradients that needed to be computed for training an LDM without LoRA still need to be computed in order to propagate backwards. The new weight matrices also need gradients, which represents an overhead that is captured by longer training times.

---

### Official Review · Reviewer_Tmmk · 2023-11-06

**Soundness:** 3 good
**Presentation:** 2 fair
**Contribution:** 2 fair
**Rating:** 8
**Confidence:** 4

**Summary:**

The paper presents DP-LDM, a differentially private latent diffusion model for generating high-quality, high-dimensional images. The authors build upon the LDM model (Rombach et al., 2022), and propose to pre-train the LDM with public data and fine-tune part of the model on private data. They evaluate DP-LDP on several datasets and report promising results compared with the prior SOTAs.

**Strengths:**

- The authors identify the difficulty of scaling up DP DMs and propose a parameter-efficient approach targeting at the issue.
- The authors claim new SOTA results for generating high-dimensional DP images, including those conditioned on text prompts, which is new.

**Weaknesses:**

I have no major complaints about the paper. The paper reads incremental, but it does a decent job in finding the correct hammer, which does lead to promising results. I list below some issues where the authors can improve on. I'll consider raising my score if the authors can properly address them.

1. Missing references:
  - On the model architecture: U-Net [1], transformers [2].
  - On the dataset: CelebA-HQ [3]
  - On the properties of DP: [4]
  - On privacy preserving data synthesis: [5]
2. Evaluation
  - The authors presented the accuracy results only on CIFAR-10 (in Table 2). All the remaining results on other datasets (Tables 3,4,5) are for FID. However, FID can at most be regarded as a fidelity metric, serving as a proxy for the utility of the synthetic data. It would be most straightforward to directly present the utility results, i.e., accuracy on the classification task. Can the authors add the accuracy results on CelebA for better interpretation of the results? (nit: "we see a significant drop in accuracy as shown in Table 3." -- Table 3 is about FID.)
  - Ghalebikesabi et al. [6] evaluated the high-dimensional medical dataset camelyon17, and so does the recent [7]. Have the authors considered performing evaluation on this dataset?
  - The baseline methods do not come with a cite. Is DP-diffusion [6] or [8]? Appendix A.2 suggests [6], but the caption of Fig. 8 suggests [8], which is confusing.
3. Clarity can be improved:
 - "inserted into the layers of the underlying UNet backbone": exactly where?
 - "It modifies stochastic gradient descent (SGD) by adding an appropriate amount of noise by employing the Gaussian mechanism to the gradients"
 - "However, Lin et al. (2023) do privatize diffusion" -> do not
 - "encoder $\Delta$ and decoder $\nabla$": non-standard notations



**References**

[1] Ronneberger, Olaf, Philipp Fischer, and Thomas Brox. "U-net: Convolutional networks for biomedical image segmentation." Medical Image Computing and Computer-Assisted Intervention–MICCAI 2015: 18th International Conference, Munich, Germany, October 5-9, 2015, Proceedings, Part III 18. Springer International Publishing, 2015.

[2] Vaswani, Ashish, et al. "Attention is all you need." Advances in neural information processing systems 30 (2017).

[3] Karras, Tero, et al. "Progressive Growing of GANs for Improved Quality, Stability, and Variation." International Conference on Learning Representations. 2018.

[4] Dwork, Cynthia, and Aaron Roth. "The algorithmic foundations of differential privacy." Foundations and Trends® in Theoretical Computer Science 9.3–4 (2014): 211-407.

[5] Y. Hu, et al., "SoK: Privacy-Preserving Data Synthesis," in 2024 IEEE Symposium on Security and Privacy (SP), San Francisco, CA, USA, 2024 pp. 2-2.

[6] Ghalebikesabi, Sahra, et al. "Differentially private diffusion models generate useful synthetic images." arXiv preprint arXiv:2302.13861 (2023).

[7] Lin, Zinan, et al. "Differentially Private Synthetic Data via Foundation Model APIs 1: Images." arXiv preprint arXiv:2305.15560 (2023).

[8] Dockhorn, Tim, et al. "Differentially private diffusion models." arXiv preprint arXiv:2210.09929 (2022).

**Questions:**

See the evaluation part in weaknesses

---

> ### Author Response · Authors · 2023-11-18
>
> We thank the reviewer for the positive and constructive feedback on our paper. Thanks for pointing out the missing references and further improvements we can make to make the paper clearer; we will add those to our final version.
>
> Regarding extra evaluations suggested by the reviewer:
> 1. We’re currently working on implementing the experimental setups for testing classification accuracy of the synthetic CelebA64 dataset. We will upload the results here once it is done.
> 2. We agree with the reviewer that evaluating our method on a scenario with an extreme domain gap between the public and private datasets. Hence, we are working on testing our method on the Camelyon17 dataset.
>
> Answers to each point below:
> - The baseline methods do not come with a cite. Is DP-diffusion [6] or [8]? Appendix A.2 suggests [6], but the caption of Fig. 8 suggests [8], which is confusing. : Thanks! We have corrected the typo.
>
> - "inserted into the layers of the underlying UNet backbone": exactly where? The original Unet does not contain attention modules, we followed the way of LDM to insert attention modules into layers as the way illustrated in Figure 1 (marked in red).  For input_blocks, attention modules are added to the Resnet block except the first in block and downsampling ones; For middle block, attention module is added between two Resnet blocks; For out_blocks, attention modules are added except the out block.
>
> - "It modifies stochastic gradient descent (SGD) by adding an appropriate amount of noise by employing the Gaussian mechanism to the gradients" We have clarified this sentence.
>
> - "However, Lin et al. (2023) do privatize diffusion" -> do not: Thanks, we have fixed this.
>
> - "encoder “ Δ and decoder ∇ ": non-standard notations: We understand that E and D are commonly used for encoders and decoders. Unfortunately, we have used D to note dataset to define differential privacy. We are considering changing encoder  Δ and decoder ∇  to be Enc and Dec as used in [9, 10, 11]. But if the reviewer has a suggestion on which notation we should use for encoders and decoders, we’re open to it.
>
> [9] ABL Larsen et al, Autoencoding beyond pixels using a learned similarity metric,  PMLR, 2015
>
> [10] Liu, Yihong et al, On the Copying Problem of Unsupervised NMT: A Training Schedule with a Language Discriminator Loss, ACL, 2023
>
> [11] Sprinks, Graham, and Moens, Marie-Francine, Generating Continuous Representations of Medical Texts, NAACL, 2018

---

> > ### Comment · Reviewer_Tmmk · 2023-11-22
> >
> > Thanks for the response and the additional experimental results. I have raised my score accordingly and will vote for acceptance.

---

> > > ### Author Response · Authors · 2023-11-23
> > >
> > > Thanks for increasing the score! We are happy that the reviewer is satisfied with our reply!

---

> ### Author Response · Authors · 2023-11-23
> **Conditional CelebA Results**
>
> As requested, we applied our model to conditional CelebA generation at 64x64 resolution. We began with an LDM pretrained on conditional ImageNet at the same resolution (64x64), and then fine-tuned it on CelebA where the (binary) class labels were given by the “Male” attribute. A ResNet-9 [2] classifier was trained on 50000 synthetic images, and then a test accuracy was computed on the test split of the real data. Note that these test set images were not seen in the training of the LDM or the classifier. As a baseline for comparison, we also ran publicly available DP-MEPF [1] code and computed test accuracies in a similar manner. Our results are summarized in the table below, which compares downstream test accuracies between DP-LDM and DP-MEPF [1] at several epsilon levels. All accuracies are computed as the average of three runs with standard deviations provided in parentheses. We see that our method outperforms DP-MEPF [1] at each tested epsilon level.
>
> |               | Test Accuracy |            |
> |---------------|---------------|------------|
> |               | DP-LDM (Ours) | DP-MEPF    |
> | $\epsilon=1$  | **94.5** (0.0)    | 82.9 (2.1) |
> | $\epsilon=5$  | **96.0** (0.0)    | 93.7 (0.0) |
> | $\epsilon=10$ | **96.4** (0.0)    | 93.9 (0.0) |
>
> [1] Harder, Frederik et al, Pre-trained perceptual features improve differentially private image generation, TMLR, 2023
>
> [2] K. He, X. Zhang, S. Ren and J. Sun, "Deep Residual Learning for Image Recognition," 2016 IEEE Conference on Computer Vision and Pattern Recognition (CVPR), Las Vegas, NV, USA, 2016, pp. 770-778, doi: 10.1109/CVPR.2016.90.

---

### Author Response · Authors · 2023-11-23

We extend our sincere gratitude to AC and SAC for handling our paper. We also extend our gratitude to the reviewers for their meticulous and constructive feedback, which has offered valuable insights to us. As a final note, we would like to re-emphasize the main contribution of this work.

Classical approaches for differentially private data generation typically assume a certain class of pre-specified purposes on how the synthetic data can be used. Thanks to the advance in deep generative modelling, we now can generate data for more general purpose (not sticking to the original intended use). There was a time when generative adversarial networks (GANs) achieved state-of-the-art performance in generative modelling. Then it was natural to think of privatizing GANs to generate data with differential privacy. Currently, diffusion based models achieve state-of-the-art performance in generative modelling. Before us, there were only two papers that attempted to make diffusion models differentially private. DP-DM performs well on relatively simple datasets such as MNIST and FashionMNIST, but does not perform well on more complex datasets such as CIFAR10 and CelebA32 and 64. This method also requires a long training time as a diffusion model is trained from scratch. DP-Diffusion attempts to overcome this issue by pretraining a diffusion model on using the abundance of public data and then fine-tuning on private data. However, this fine-tuning is done on all of the parameters of the diffusion model and is still computationally costly.

Our method incorporates the benefits of these two attempts. In particular, we also use diffusion models, as DMs can model complex image distributions very well. We also use public data to reduce the training time (which leads to a smaller privacy loss). On top of these ingredients, we incorporate the latent diffusion models that reduce the dimension of diffusion process and selectively fine-tune attention modules in the LDMs to further reduce the fine-tuning time and dimensionality of the parameters to be fine-tuned. We provided ablation studies on which layer's attention modules to be fine-tuned for better privacy-accuracy trade-offs. We found generally fine-tuning those in Out-blocks was more helpful than those in In-blocks.

Thanks to the reviewers, we also tested LoRA to further reduce the number of parameters to fine-tune, where we found that LoRA was not particularly useful and our current DP-LDM still outperforms the LoRA implementation, thanks to the use of large batch size in our algorithm. However, we believe thinking of the model's intrinsic dimensionality and its structure seems an intriguing direction when it comes to fine-tuning a model.

When it comes to the performance, our DP-LDM algorithm achieves state-of-the-art results in image generation with differential privacy. Additionally, thanks to our particular model choice, we can furthermore generate realistic images at the scale of 256x256 and text-to-image generation with differential privacy, which was not explored before, and which we hope the reviewers and AC/SAC to regard as making a leap in DP data generation.

Therefore, we respectfully request recognition of the contributions we make in advancing differentially private data generation research.

Thanks again for considering our work.

---

### Public Comment · ~Yizhe_Li1 · 2023-12-12
**Some Concerning Comments**

Thanks for the paper and great contribution towards the DPML community. Here I have some follow-up questions that would be great if you could answer.
### LoRA-Related:
* When using LDM with LORA [1] (provided in Hugging Face with rank 8) as finetuning technique for the finetuning of text2image models, the speed of LORA is clearly faster and more efficent then DP-LDM (around 1.5 times faster). Does the author compared the tuning time with other models under various parameter counts?
* Can the LORA experiment be more detailedly describe, such as which part was applied LORA for the fine-tuning process? It is hard to set a definitive conclusion that DP-LDM exceed LORA without comparing under sufficient configuration.
* In the supplemental experiment of LoRA, it can be seen that LDM with LoRa achieves a lesser performance than DP-LDM, this might be relevant to the size of the model, it might be more comprehensive to see if using models with larger amount of parameters might make a difference?
### Experiment-Related:
* Since the LAION-400M contains tons of facial record with high quality, simply claiming that DP-LDM achieves great FID on the facial dataset might not be strong enough.
* Does the 'freeze the BERT embedder' refers to freezing all the conditioning embedder?
* It would be great if the authors could open-source the checkpoint for replication so as to mitigate the above concerns.

[1]: https://github.com/huggingface/peft

---

### Meta-Review · Area_Chair_nVZ6 · 2023-12-05

**Metareview:**

The paper studies the problem of differentially privately training latent diffusion models. It proposes a training recipe where we start from a latent diffusion model that was pre-trained on public data, and then use DP-SGD to fine-tune, on a private/sensitive dataset, the attention modules and the embedders for conditioned generation. The paper then carries out an experimental evaluation showing that the proposed method has a superior privacy-utility trade-off compared to prior work.

The problem studied in this paper is important and timely. However, the novelty of the paper is limited. The method is a quite direct application of known techniques. In its present form, the paper does not meet the novelty bar for publication at ICLR.

**Justification For Why Not Higher Score:**

The paper's novelty is quite limited.

**Justification For Why Not Lower Score:**

n/a

---

### Decision · Program_Chairs · 2024-01-16

Reject